# Innate immune evasion revealed in a colorectal zebrafish xenograft model

Vanda Póvoa[1], Cátia Rebelo de Almeida[1], Mariana Maia-Gil [1], Daniel Sobral[2], Micaela Domingues [1], Mayra Martinez-Lopez [1], Miguel de Almeida Fuzeta[1], Carlos Silva[1], Ana Rita Grosso [2] & Rita Fior [1✉]

Cancer immunoediting is a dynamic process of crosstalk between tumor cells and the immune system. Herein, we explore the fast zebrafish xenograft model to investigate the innate immune contribution to this process. Using multiple breast and colorectal cancer cell lines and zAvatars, we find that some are cleared (regressors) while others engraft (progressors) in zebrafish xenografts. We focus on two human colorectal cancer cells derived from the same patient that show contrasting engraftment/clearance profiles. Using polyclonal xenografts to mimic intra-tumor heterogeneity, we demonstrate that SW620_progressors can block clearance of SW480_regressors. SW480_regressors recruit macrophages and neutrophils more efficiently than SW620_progressors; SW620_progressors however, modulate macrophages towards a pro-tumoral phenotype. Genetic and chemical suppression of myeloid cells indicates that macrophages and neutrophils play a crucial role in clearance. Single-cell-transcriptome analysis shows a fast subclonal selection, with clearance of regressor subclones associated with IFN/Notch signaling and escaper-expanded subclones with enrichment of IL10 pathway. Overall, our work opens the possibility of using zebrafish xenografts as living biomarkers of the tumor microenvironment.

[1] Champalimaud Centre for the Unknown, Champalimaud Research, Champalimaud Foundation, Lisbon, Portugal. [2] UCIBIO, Departamento de Ciências da Vida, Faculdade de Ciências e Tecnologia, Universidade NOVA de Lisboa, Caparica, Portugal. ✉email: rita.fior@research.fchampalimaud.org

Clinically detectable tumors represent the ultimate consequence of tumor immunoediting, which includes the detection and clearance of the majority of the immunogenic clones by the immune system[1]. Clones that escape immune detection further hijack immune cells to support tumorigenesis[2]. Although the concept of immunoediting is well established, the role of innate immune cells in shaping and selecting subclones, as well as the mechanisms that allow for innate immune evasion remain less explored[3].

In recent years, there has been a major effort to uncover the role of adaptive immunity on tumor immune surveillance/evasion/suppression, which has been translated into promising new immunotherapies[4]. Immune checkpoint therapies aim to remove inhibitory pathways that block anti-tumor T-cell responses in the tumor microenvironment (TME). However, therapy may fail because tumor cells do not express sufficient neo-antigens (not immunogenic enough)[5]. Another major obstacle may be the presence of a suppressive (cold) TME composed of stroma and a variety of immune cells, such as regulatory T cells (Treg), myeloid-derived suppressor cells (MDSC), alternative activated pro-tumoral macrophages ("M2-like"), and neutrophils ("N2-like"), that may block anti-tumor immune responses[5,6]. In fact, innate myeloid-derived cells effectively represent the major component of the TME in most solid tumors[5,7], often outweighing lymphocytes or even the tumor cells themselves. These populations of the immune system are present in all tissues. However, their role in cancer-induced immune suppression and immunotherapy remains less explored and understood. Increasing evidence supports a crucial anti- and pro-tumorigenic role for innate immune cells[4]. Importantly, innate pro-tumorigenic states are highly dynamic and can be selectively reverted[8], creating the possibility for new and more effective therapeutic approaches.

The zebrafish model has emerged as a powerful tool to study tumor biology and interactions with the immune system[9–12]. Zebrafish have a highly conserved vertebrate innate immune system, including complement, Toll-like receptors, neutrophils, and macrophages capable of phagocytic activity. Another advantage is that the full maturation of adaptive immunity only occurs at 2–3 weeks post-fertilization[13,14]. This offers a time window to study exclusively innate immune response in vivo, independent of the adaptive system. In addition, transparency allows for unprecedented real-time imaging of cell–cell interactions and genetic tractability enables the engineering of reporter lines and mutants[14].

Recently, we have optimized zebrafish patient-derived xenografts (zPDXs)—"zAvatars" for personalized medicine[15]. The assay is based on the injection of labeled tumor cells into zebrafish embryos for assessment of tumor behavior and response to therapy in just 4 days. Although only innate immunity is active, we observed some heterogeneity in tumor engraftment. Here we hypothesize that the zebrafish innate immune system can be modulated by the tumor; itself capable of generating an immunosuppressive TME or subjected to elimination. In the present study, we apply a combination of zebrafish mono- and polyclonal xenografts, "zAvatars", zebrafish mutants and transgenics, mouse xenografts, re-transplantation experiments, and single-cell transcriptomics to test this hypothesis. We focus on a pair of human colorectal cancer (CRC) cells derived from the same patient at different stages of tumor progression: SW480 was derived from the primary tumor, and SW620 from a lymph node metastasis isolated 6 months later. SW480_regressors engraft poorly and most tumors are cleared during the 4 days of the assay, whereas SW620_progressors engraft very efficiently. Mixing SW480_regressors with SW620_progressors in polyclonal tumors reduces clearance of SW480_regressors, suggesting that progressors can induce an immune suppressive environment. Indeed, not only

SW620 progressor tumors recruit less neutrophils and macrophages to the TME, but also polarize macrophages toward a M2-like pro-tumoral phenotype. In addition, MIX polyclonal tumors show an immune profile similar to the progressor tumors, i.e., reduced numbers of innate cells and "M2-like" polarization. Genetic and chemical depletion of myeloid cells confirms that macrophages and neutrophils play a crucial role in this clearance process. To test whether innate immunoediting is occurring in this short time frame, we perform re-transplantation experiments of SW480 escaper tumors and show that these tumors engraft more efficiently and generate bigger tumors with reduced macrophage infiltration. Finally, single-cell RNA-seq reveals the in vivo clearance and expansion of specific subclones.

## Results

**Zebrafish xenografts display differential engraftment profiles.** Using a zebrafish xenograft model[15], we investigated the engraftment efficiency of multiple human breast and CRC cell lines. At 4 days post injection (4 dpi), we found that different cancer cell lines display distinct engraftment profiles in zebrafish xenografts. Of note, we describe engraftment as the frequency of xenografts that present a tumor (at least 30 tumor cells) at 4 dpi (Fig. 1a) and clearance as engraftment inhibition. We observed that some cancer cell lines present a high engraftment rate—above 80%, while others engraft poorly with an average engraftment rate of ~20–30%. We define here these tumors as progressors and regressors, respectively, following Schreiber nomenclature[16] (Fig. 1a).

Strikingly, we observed differences in engraftment profiles between cancer cells derived from the same patient at different stages of tumor progression. While SW480 cells derived from the primary tumor present a regressor behavior, SW620 cells isolated from a lymph node metastasis 6 months later[17,18] show a progressor phenotype (Fig. 1a). These differences in engraftment rates between both tumor cells were also originally reported in mouse xenografts[18].

Importantly, engraftment/clearance capacity did not seem to correlate to proliferation potential or basal cell death. This is exemplified by the breast cancer cells Hs578T_progressors, which display a high engraftment rate (~95% engraftment), despite their low proliferation and high level of apoptosis in comparison, for instance, with breast cancer MDA-MB-468, which display lower engraftment but are more proliferative and less apoptotic (Supplementary Fig. 1a–d). Also, although SW620_progressors are highly proliferative compared with SW480_regressors, SW620_progressors present higher levels of apoptosis (Supplementary Fig. 1e–h).

Moreover, paradoxically, we observed that SW480_regressors upon chemo- (FOLFOX-FO) or radiotherapy (RAD), may increase their engraftment rate, and this can also be observed in patient-derived xenografts (zAvatars) (Fig. 1b). Given the fact that chemo/radiotherapy may elicit an immunosuppressive effect, we hypothesized that this could reduce the zebrafish host anti-tumor response, originally responsible for the regressor (clearance) behavior.

**Transcriptomic analysis between SW480 and SW620 xenografts.** We next performed a general comparative transcriptomic analysis between SW480_regressors and SW620_progressors. We sought to focus on the SW480/SW620 pair of cell lines since they derived from the same patient and therefore illustrate intra-patient heterogeneity and eventually the original immunoediting process from primary to metastasis progression. To this end, SW480 and SW620 tumors were dissected from zebrafish xenografts at 2 dpi, a timepoint that corresponds to the timing when

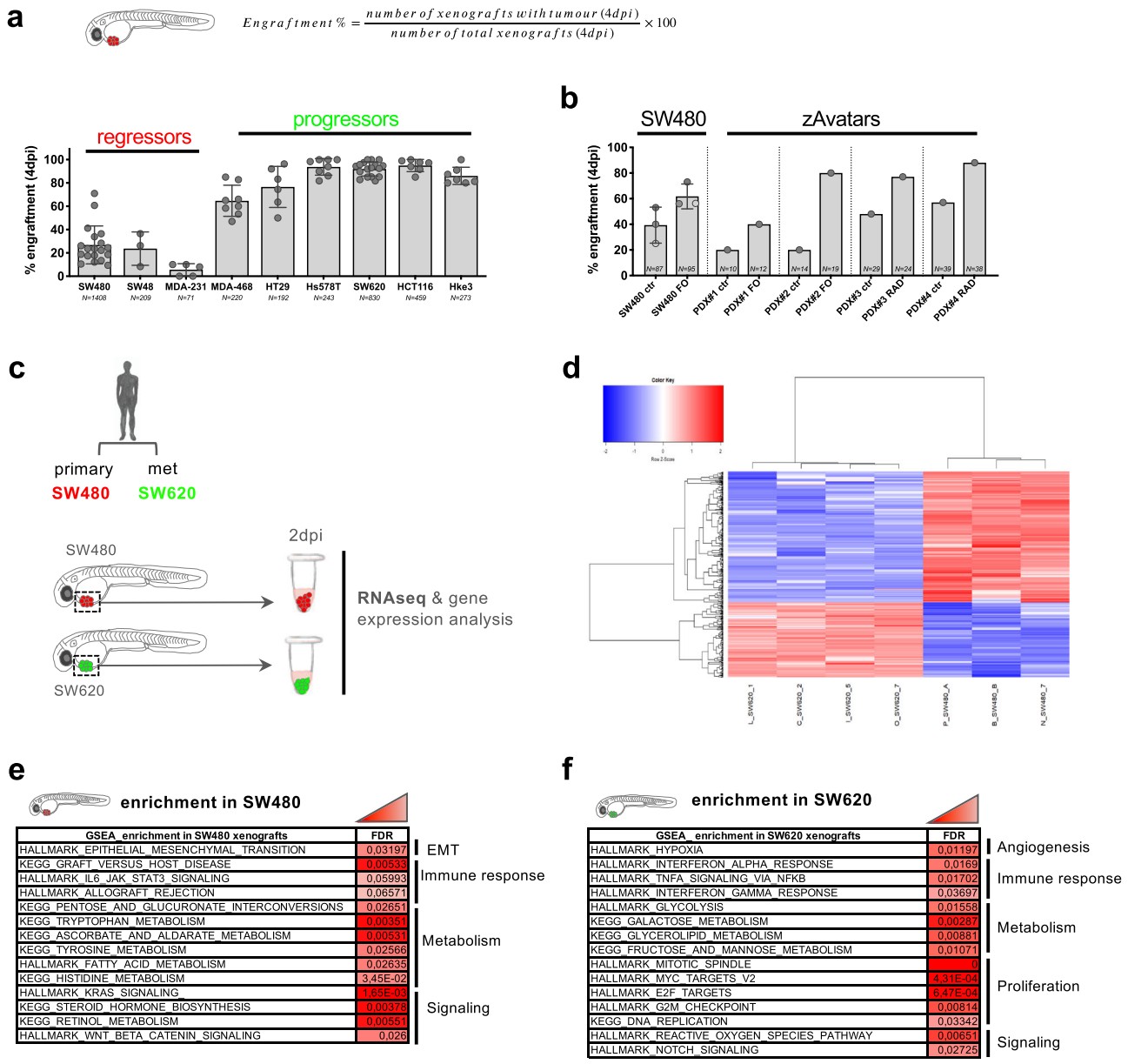

**Fig. 1 Human cancer cells display differential engraftment profiles in zebrafish. a** Engraftment is the ratio between the number of zebrafish xenografts that maintain a tumor at 4 days post injection (dpi) and the number of total xenografts that were originally successful injected and survived until day 4. MDA-MB-231 (MDA-231), MDA-MB-468 (MDA-468), and Hs578T are breast cancer cell lines. SW480, SW48, HT29, SW620, HCT116, and Hke3 are colorectal (CRC) cancer cell lines. Tumor cells were labeled and injected into the perivitelline space (PVS) of 2 days post-fertilization (dpf) zebrafish embryos. Each dot represents one independent experiment, number of independent experiments: 19 SW480, 3 SW48, 5 MDA-231 12 MDA-468, 5 HT29, 7 Hs578T, 22 SW620, 6 HCT116, 7 Hke3. Total number of xenografts analyzed (N) is depicted in the charts. Error bars indicate mean ± S.D. **b** Engraftment of SW480 and zebrafish patient-derived xenografts (zPDX-zAvatars) at 4 dpi, treated with FOLFOX (FO) and radiotherapy (RAD) and their respective controls. Each dot represents one independent experiment (3 SW480, 1 zPDX). Total number of xenografts analyzed (N) is depicted in the charts. Error bars indicate mean ± S.D. See also Supplementary Fig. 1. **c–f** Comparative transcriptomic analysis between SW480 and SW620 xenografts. **c** Schematic representation of the experiment where SW480 (in red) and SW620 (in green) tumors were dissected at 2 dpi for RNA extraction (~30 tumors of each condition). **d** Heatmap presents a two-dimensional dendogram (based on Pearson's correlation coefficient distance) of log2 counts-per-million (logCPM), normalized expression values of differentially expressed genes (N = 459, cut-off of FDR < 0.05 and absolute log2FC > 1) in SW480 (low engraftment) versus SW620 (high engraftment) comparison, where colors represent expression values scaled by row (Z-scores). **e**, **f** GSEA of SW480 and SW620 xenografts. Source data are provided as a Source data file.

clearance is actively taking place (Supplementary Fig. 2). A pool of ~30 tumors from at least three independent experiments was collected for RNA extraction (Fig. 1c). The remaining xenografts from the same experiments were followed until 4 dpi to determine final engraftment rates. We only used RNA samples from experiments where SW480 engraftment was lower than ~30% and SW620 engraftment was higher than 90%.

A differential expression analysis revealed 459 differentially expressed genes (DEGs) between the two types of xenografts (Fig. 1d, Supplementary Data 1). A gene set enrichment analysis

(GSEA)[19] revealed an enrichment mainly in three biological processes: immune response, metabolism and signaling (Fig. 1e, f). Whereas genes involved in epithelial to mesenchymal transition (EMT) were specifically enriched in SW480 xenografts, genes involved in proliferation and hypoxia/angiogenesis were specifically represented in SW620 tumors (Fig. 1e, f). The enrichment analysis is in accordance with our earlier results, where SW480 showed an increased metastatic potential and SW620 an increased capacity to recruit blood vessels[15]. We also identified several immune-related pathways in SW480 enriched DEGs, in particular those involved in graft-versus-host disease, IL6 signaling, and allograft rejection pathways. In contrast, SW620 DEGs were characterized by an enrichment in IFN and TNF signaling, ROS and NOTCH pathways (*Jagged1*, *MAML2*), but not in graft-versus-host disease or allograft rejection pathways (Fig. 1e, f). These results suggest that SW480 tumors express signals that may stimulate clearance, while SW620 tumors have reduced activity of rejection-related pathways.

**Progressors are able to protect regressors from being cleared**. In order to test if progressors were able to induce a suppressive environment, and thus avoid clearance of regressors, SW620_progressors were mixed with SW480_regressors, thereby generating polyclonal xenografts in vivo. To distinguish the two cell lines, SW480 cells were labeled with red CM-DiI-dye and SW620 with green CMFDA-dye and mixed in a 1:1 proportion (Fig. 2a, b). The three conditions were tested in parallel—SW480 (red) alone, SW620 (green) alone, and MIX (SW480 + SW620) and engraftment quantified at 4 dpi. As expected, SW480 cells presented a low average engraftment rate of ~20%, with the majority of tumors being cleared from the zebrafish host; whereas SW620 had an average engraftment rate of ~90%. However, when mixed, engraftment of SW480 increased to more than double, to ~60% (P < 0.0001). In contrast, engraftment of SW620 decreased ~35% (P < 0.0001) in relation to when it is alone (Fig. 2c). Analysis of the relative proportions of each clone in each MIX xenograft, by confocal microscopy, showed that both populations were always present, with SW620 behaving as the dominant clone, making up ~70% of the tumor (Fig. 2d). Interestingly, when we compared the size of each population (number of cells) we found that the number of SW480 cells increased in polyclonal xenografts, i.e., SW480 benefits from the proximity of SW620 cells, suggesting that SW620 cells can protect SW480 cells from clearance and possibly provide some survival cues (Fig. 2e).

Next, we engrafted a mixture of SW480_regressor with another CRC progressor cell line derived from a different patient—HCT116 (Fig. 2f, g). In this instance, in the presence of HCT116, the engraftment rate of SW480 was further increased, from ~20 to 90% (P < 0.0001) (Fig. 2h), while analysis of each xenograft revealed SW480:HCT116 frequencies of 30:70% (Fig. 2i). Once again, the size of SW480 tumors increased in the presence of a progressor tumor cell (Fig. 2j).

These results suggest that "regressors" can indeed lose their "regression" profile in the presence of "progressors" and that the latter might generate a protective immunosuppressive microenvironment. These results are in accordance with mouse xenograft studies, which show that advanced metastatic tumors engraft more efficiently and are more immunosuppressive than primary tumors[20,21].

**SW480 regressor TME is enriched in innate immune cells**. To evaluate if regressors and progressors are able to generate different tumor ecosystems, we analyzed the presence of neutrophils and macrophages in the tumors, the main innate immune cells present at this stage of development (2–6 days post fertilization-

dpf)[14,22]. To this end, we injected SW480, SW620, and MIX tumor cells into *Tg(mpx:eGFP)*[23] and *Tg(mpeg1:mcherry-F)*[24] zebrafish hosts, which have neutrophils (Fig. 3a, b) and macrophages (Fig. 3e, f) labeled, respectively.

As early as 24 hpi (1 dpi), we could detect a significant higher recruitment of neutrophils and macrophages to the SW480 tumors in comparison to SW620 (neutrophils P < 0.0001, macrophages P = 0.0011), a difference that was maintained and reinforced at 4 dpi (neutrophils P < 0.0001, macrophages P = 0.0089) (Fig. 3c, d, g, h). Interestingly, MIX tumors showed a TME similar to SW620, with significant lower recruitment of neutrophils and macrophages than SW480 tumors (Fig. 3, SW480 vs MIX neutrophils $P_{4dpi} < 0.0001$, macrophages $P_{4dpi} = 0.0025$). These results suggest that the presence of SW620 in the MIX is able to block the recruitment of immune cells toward the tumor. We next questioned whether immune cell recruitment was associated with the total number of tumor cells within the tumoral mass. Linear regression analysis of the tumor size vs immune cell counts suggests a weak correlation between tumor size and immune cell infiltrates in SW480 tumors, but moderate in SW620 tumors (Supplementary Fig. 3).

**SW480 and SW620 tumors modulate zebrafish macrophage polarization**. In the TME, tumor-associated macrophages (TAMs) and neutrophils (TANs) can either adopt an anti-(M1/N1-like) or pro-tumoral (M2/N2-like) phenotype, known to be modulated by multiple tumor-derived signals[24,25]. To investigate the polarization state of macrophages in both TMEs, SW480, and SW620 cells were injected into double transgenic animals *Tg (mpeg1:mCherry-F; tnfa:eGFP-F)*[24] and each population was analyzed at 1 and 4 dpi (Fig. 4a, b and Supplementary Fig. 4a, b). Quantification of the immune cell populations showed that SW480_regressors are able to recruit a significantly higher number of inflammatory cells (TNFa positive cells and M1-like TNFa+mpeg+), than SW620_progressors, since 1 dpi (Supplementary Fig. 4a, b, M1-like $P_{1dpi} = 0.0003$; $P_{4dpi} = 0.001$). Moreover, when we analyzed the proportions of M1-like (TNFa+) versus M2-like (TNFa−) macrophages, we observed that the SW480 TME presented ~57% M1-like to 43% M2-like- macrophages at 4 dpi (Fig. 4c, d). In clear contrast, the TME of SW620_progressors cells presented a ratio of ~35% M1-like to ~65% M2-like macrophages (Fig. 4c). Interestingly, a progressive increase in M2-like-(TNFa−) macrophages in the TME of SW620 from 1 to 4 dpi was observed (Fig. 4c). This result suggests that SW620_progressor cells can polarize macrophages to a M2-like pro-tumoral state. In addition, the MIX xenografts again show similar dynamics to SW620 xenografts (M2- > M1-like macrophages), from 1 to 4 dpi (Fig. 4c, d).

Moreover, as expected, we detected a higher phagocytic activity (displayed by M1-like TNFa+/mpeg+ and TNFa+/mpeg− cells) in SW480 TME than in SW620 (Supplementary Fig. 4c–i, P < 0.0001). In summary, these results show that human tumor cells are able to modulate the zebrafish TME toward a more anti- or pro-tumoral state, through macrophage polarization and consequent phagocytic properties.

**Engraftment of MIX tumors correlates with SW620 ratio**. Next, we questioned whether clonal proportions could affect tumor engraftment and TME modulation. Zebrafish embryos were injected with mixtures of regressors with progressors at different ratios (1:3, 1:1, and 3:1) (Supplementary Fig. 5a). We found that a proportional increase in the number of SW620_progressors cells in polyclonal tumors correlates with higher engraftment rates (Supplementary Fig. 5b, c, R2 = 0.78, P < 0.0001). Interestingly, instead of a steady reduction of the immune infiltrate into the

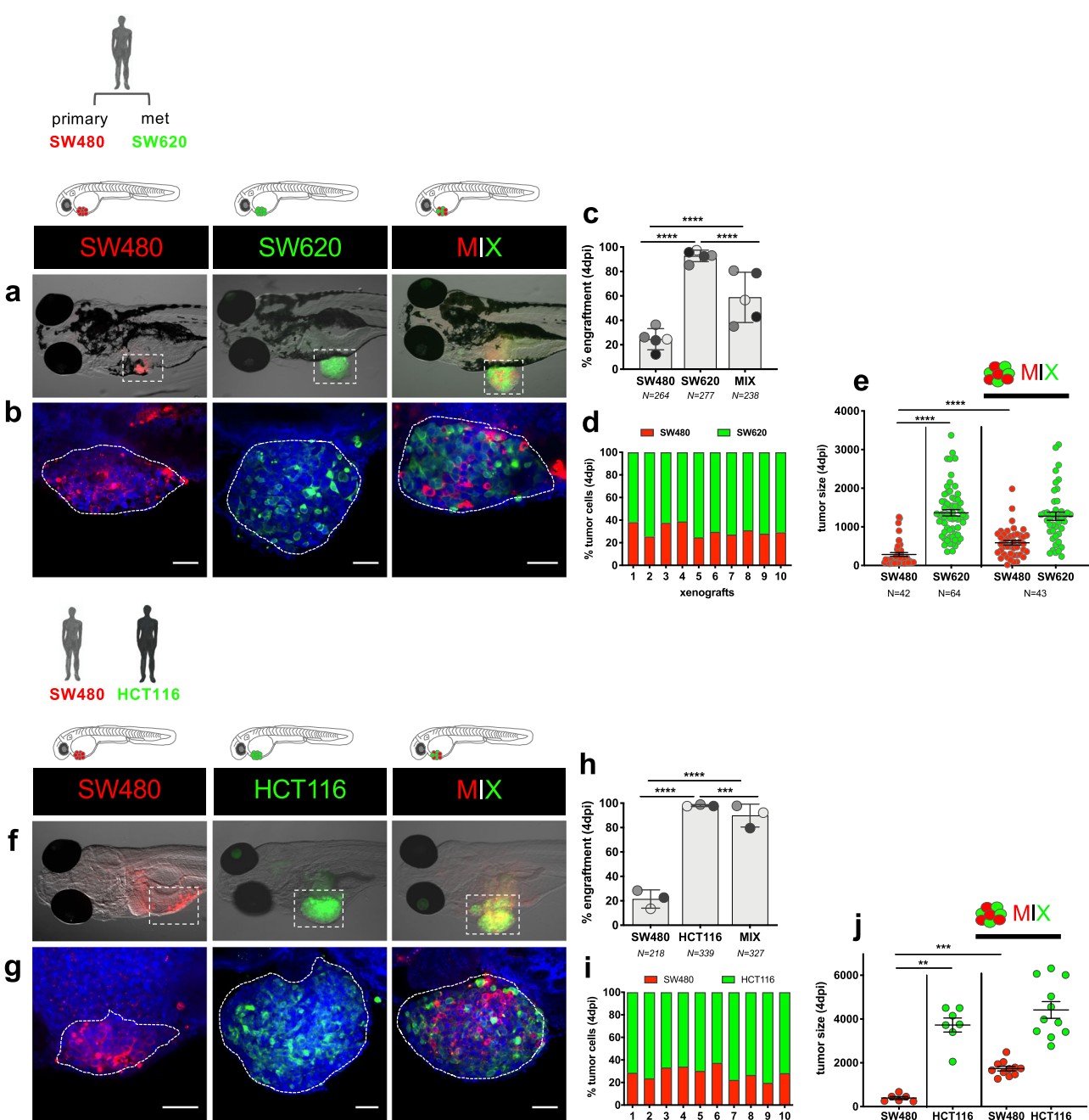

**Fig. 2 Progressor tumor cells are able to protect regressors from being cleared.** Tumor cells were labeled with lipophilic dyes and injected into the PVS of 2dpf zebrafish embryos. **a**, **b** Representative images of SW480 (in red), SW620 (in green), and MIX (1:1) polyclonal zebrafish xenografts at 4 dpi. **a** Fluorescence stereoscope images. **b** Confocal images. **c** Engraftment quantification at 4 dpi (Fisher exact test ****$P < 0.0001$). Graph shows the mean ± S.D. Each dot represents one independent experiments (5), and each set of independent experiments is represented in a different gray color. **d** Representative quantification of the proportions of each clone within each xenograft ($N = 10$) from four independent experiment. **e** Quantification of tumor size (no. of tumor cells) at 4 dpi (unpaired two-sided Mann–Whitney test ****$P < 0.0001$). Graph shows the mean ± SEM from four independent experiments, each dot represents one xenograft. **f**, **g** Representative images of SW480 (in red), HCT116 (in green), and MIX (1:1) zebrafish xenografts at 4 dpi. **f** Fluorescence stereoscope images. **g** Confocal images. **h** Engraftment quantification at 4 dpi (Fisher exact test ****$P < 0.0001$, ***$P = 0.0005$). Graph shows the mean ± S.D. Each dot represents one independent experiment ($N = 3$), and each set of independent experiments is in a different gray color. **i** Representative quantification of the cell proportions of each clone within each xenograft ($N = 10$) from one independent experiment. **j** Quantification of tumor size (no. of tumor cells) at 4 dpi (unpaired two-sided Mann–Whitney test **$P = 0.0012$, Cohen's D $g = 4.88$; ***$P = 0.0002$, Cohen's D $g = 4.32$). Graph shows the mean ± SEM from one independent experiment, each dot represents one xenograft. Scale bars: 50 μm. Dashed lines encircle tumor areas. Nuclei are stained with DAPI (blue). $N$ is depicted in the charts. Source data are provided as a Source data file.

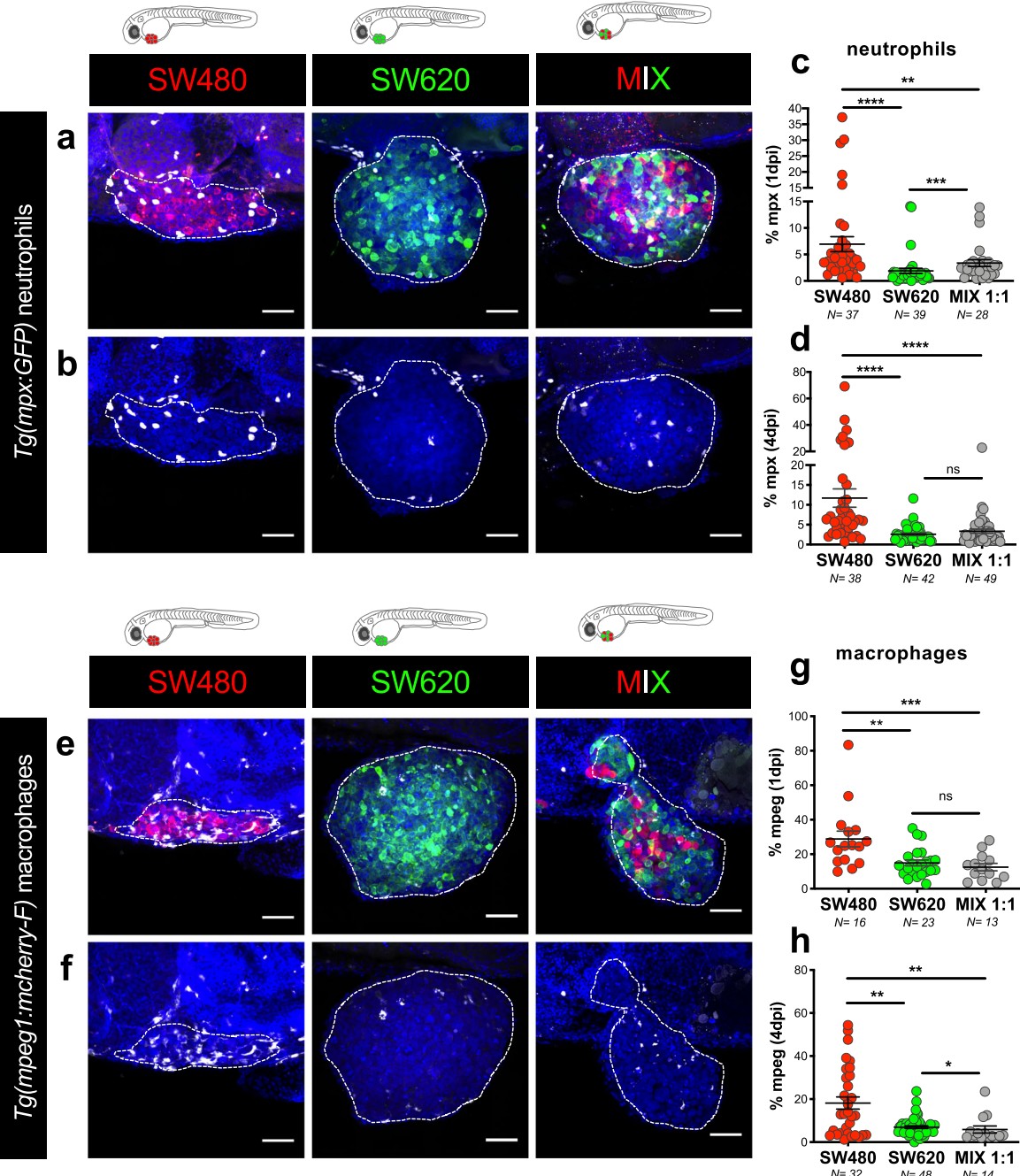

**Fig. 3 SW480_regressor TME is enriched in innate immune cells. a, b** Representative confocal projection images of neutrophils in SW480, SW620, and MIX tumors from *Tg(mpx:eGFP)* zebrafish xenografts at 4 dpi. **c, d** Quantification of neutrophils percentage (no. of neutrophils/no. of tumor cells x 100) within SW480, SW620, and MIX TME, at 1 dpi (**c**, ****$P < 0.0001$, ***$P = 0.0002$, **$P = 0.0094$) and 4 dpi (**d**, ****$P < 0.0001$, ns = 0.39). **e, f** Representative confocal projection images of macrophages in SW480, SW620, and MIX tumors from *Tg(mpeg1:mcherry-F)* zebrafish xenografts at 4 dpi. **g, h** Quantification of macrophage percentage (no. of macrophages/no. of tumor cells x 100) within SW480, SW620, and MIX tumors, at 1 dpi (**g**, ***$P = 0.0009$, **$P = 0.0011$, ns = 0.45) and 4 dpi (**h**, 480 vs 620 **$P = 0.0089$, 480 vs MIX **$P = 0.0025$, *$P = 0.024$). SW480 (red) and SW620 (green), neutrophils (white) and macrophages (white) fake colors. Scale bars: 50 μm. Dashed lines encircle tumor areas. Nuclei are stained with DAPI. *N* is depicted in the chart. Each dot represents one xenograft. Error bars indicate mean ± SEM (from three independent experiments). All data were analyzed using unpaired two-sided Mann–Whitney test. See also Supplementary Fig. 3. Source data are provided as a Source data file.

polyclonal tumors, the presence of SW620, even in a 3:1 ratio (SW480:SW620), was sufficient to block neutrophil recruitment ($P = 0.0066$) (Supplementary Fig. 5d). However, macrophage recruitment is only reduced when SW620 increases up to 50% of tumor cells (Supplementary Fig. 5e), suggesting that neutrophil and macrophage recruitment have different dynamics, possibly modulated by different mechanisms.

**Zebrafish innate immune cells regulate SW480 clearance.** The above results show that SW620_progressors protect SW480_re-gressors from being cleared and that SW480 cells are able to recruit more efficiently innate immune cells. Moreover, increas-ing amounts of SW620 in MIX xenografts correlate with increased engraftment of SW480 and the presence of SW620 seems sufficient to reduce immune cell infiltration. All

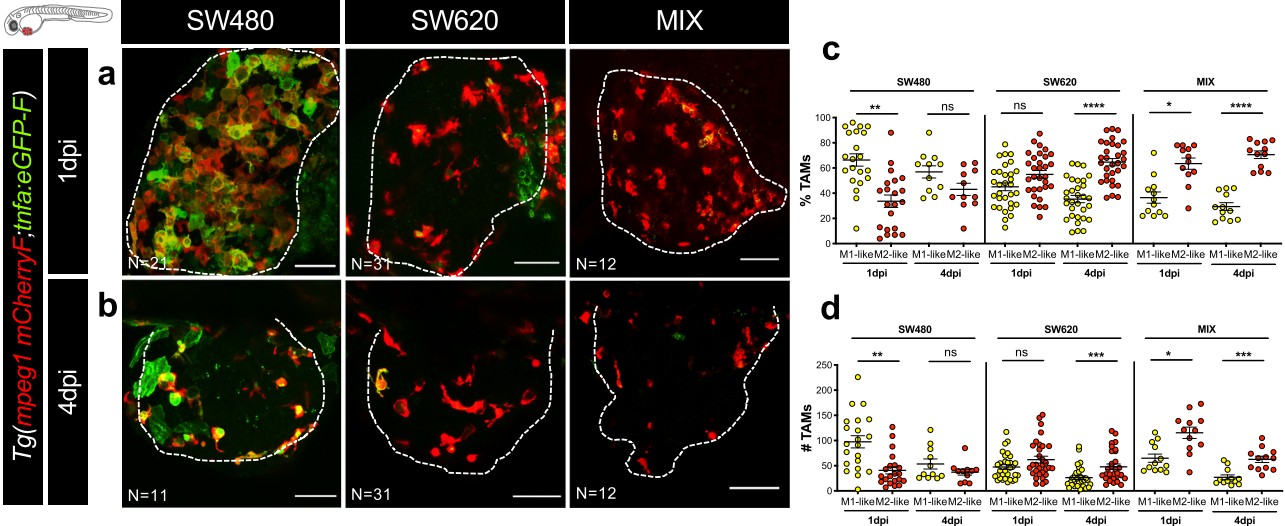

**Fig. 4 SW480 and SW620 human tumor cells modulate zebrafish macrophage polarization. a, b** Representative confocal images of SW480, SW620, and MIX xenografts injected in *Tg(mpeg1:mcherry-F, tnfa:GFP-F)* at 1 and 4 dpi. Red: macrophages; green: TNFa+ cells; yellow: overlay of macrophages in red and TNFa+ cells in green—M1-like macrophages. **c** Proportion of M1- and M2-like macrophages in the TME at 1 and 4 dpi (paired two-sided *t* test, **$P$ = 0.0033, ns = 0.1833, ns = 0.1160, ****$P$ < 0.0001, *$P$ = 0.0116, ****$P$ < 0.0001). **d** Quantification of absolute numbers of M1- and M2-like macrophages in the TME at 1 and 4 dpi (paired two-sided Wilcoxon rank test **$P$ = 0.0016, ns = 0.3086, ns = 0.1473, ***$P$ < 0.0002, *$P$ = 0.0205, ***$P$ < 0.0005). Scale bars: 50 µm. Dashed lines encircle tumor areas. $N$ is depicted in the images. In **c** and **d**, the number of xenografts analyzed is: SW480_1 dpi $N$ = 21, SW480_4 dpi $N$ = 11, SW620_1 dpi $N$ = 31, SW620_4 dpi $N$ = 31, MIX_1 dpi $N$ = 12, and MIX_4 dpi $N$ = 12. Images are maximum intensity projections. Each dot represents one xenograft. Error bars indicate mean ± SEM (from 2 independent experiments in SW480, 3 in SW620, and 1 experiment for the MIX). Source data are provided as a Source data file.

together, these results suggest that innate immunity plays an active role in clearance/engraftment.

To directly test this, we injected both CRC cell lines into mutant zebrafish embryos that have either a transient downregulation of neutrophils (*runx1^w84x* mutant)[26] or of macrophages (M-CFS receptor/fms mutant *csf1ra^j4blue panther*)[27] (Fig. 5a–b). The results show that *runx1^w84x* and *panther* mutants present a significant increase in the engraftment of SW480 regressors cells (Fig. 5c). In *runx1^w84x* mutants, we observed a significant 3.2-fold increase of engraftment ($P$ < 0.0001), whereas in *panther* mutants we observed a 2.8-fold of increase ($P$ < 0.0001) (Fig. 5c). In contrast, downregulation of neutrophils or macrophages had no significant impact on SW620_progressors' engraftment rate. Interestingly, quantification of tumor size in each background shows that SW480 regressors increase their size in *panther* mutants, which have reduced number of macrophages (Fig. 5d, $P$ = 0.0013).

Overall, our results suggest that both myeloid cells play a crucial role in the SW480_regressors' clearance and that SW620_progressors are able to evade and/or suppress the host innate immune system.

**Resident and definitive macrophages are required for SW480 clearance.** Recent studies have shown differential functions for resident macrophages and hematopoietic monocyte-derived macrophages in tumorigenesis[28–30]. In 3 dpf zebrafish larvae, macrophages are distributed in several peripheral tissues, such as the brain, heart, retina, and muscle, and in the caudal hematopoietic tissue (CHT), a transient hematopoietic tissue[9]. In *panther* mutants (*csf1ra^j4blue*), it has been shown that there is an overall ~40% reduction of the macrophage population and impairment of their migration. However, the tissue-resident macrophages (derived from the primitive and transient waves of hematopoiesis) show a stronger reduction (~60%) than macrophages derived from the second-monocytic definitive wave (CHT-20%)[31–33]. The results observed in *panther* mutants thus reflect mostly the contribution of

the resident macrophages[32]. To further investigate the role of the different macrophages in tumor clearance, we depleted most macrophage population by using Liposome-Clodronate (L-clodronate), which targets macrophages regardless of their embryonic origin. Strikingly, upon almost complete macrophage depletion (without affecting neutrophil numbers[32], see Fig. 5e–g and Supplementary Fig. 6a, b), SW480 engraftment reaches almost 100% (Fig. 5k, $P$ < 0.0001), contrasting with the significant but less pronounced engraftment increase in *panther* mutants (~60%, Fig. 5c). Moreover, quantification of the tumor size also shows that SW480 tumor size increases by almost 2-fold (Fig. 5h–j, l, L-PBS vs L-Clodro, $P$ = 0.02). In summary, our results highlight a major role for both tissue-resident and peripheral macrophages in tumor clearance.

**Conservation of phenotypes in mouse xenografts.** Zebrafish has become a relevant animal model to study cancer. This is only possible due to the major conservation of genes and signaling pathways between human and zebrafish[9,13]. Nevertheless, as mouse is the gold-standard model in cancer and immunology, we tested if the phenotypes unveiled in zebrafish were conserved across species. SW480, SW620, and MIX mouse xenografts were generated using as host the immunocompromised mice strain Rag1−/− C57BL6/N, lacking mature B and T cells[34].

However, in contrast with previous mouse xenografts[18] and our zebrafish studies using SW480 and SW620 cells, we could not detect major differences in engraftment capacity between SW480 and SW620. This discrepancy between our mouse experiments with zebrafish engraftment and the previous published mouse studies may be a reflection of our use of different mutations and background strains, both of which were immunodeficient but with different immunological repertoires. Instead of using mouse Rag1−/− C57BL6/N, Hewitt et al.[18], used BalbC nude (Foxn1 mutation) mice, which lack a thymus and functional B cells, whereas Rag1−/− mutants lack mature B and T cells[34].

Nevertheless, analysis of the F4/80+CD80+ macrophages, showed that SW480 tumors were more enriched in "M1-like" anti-tumoral

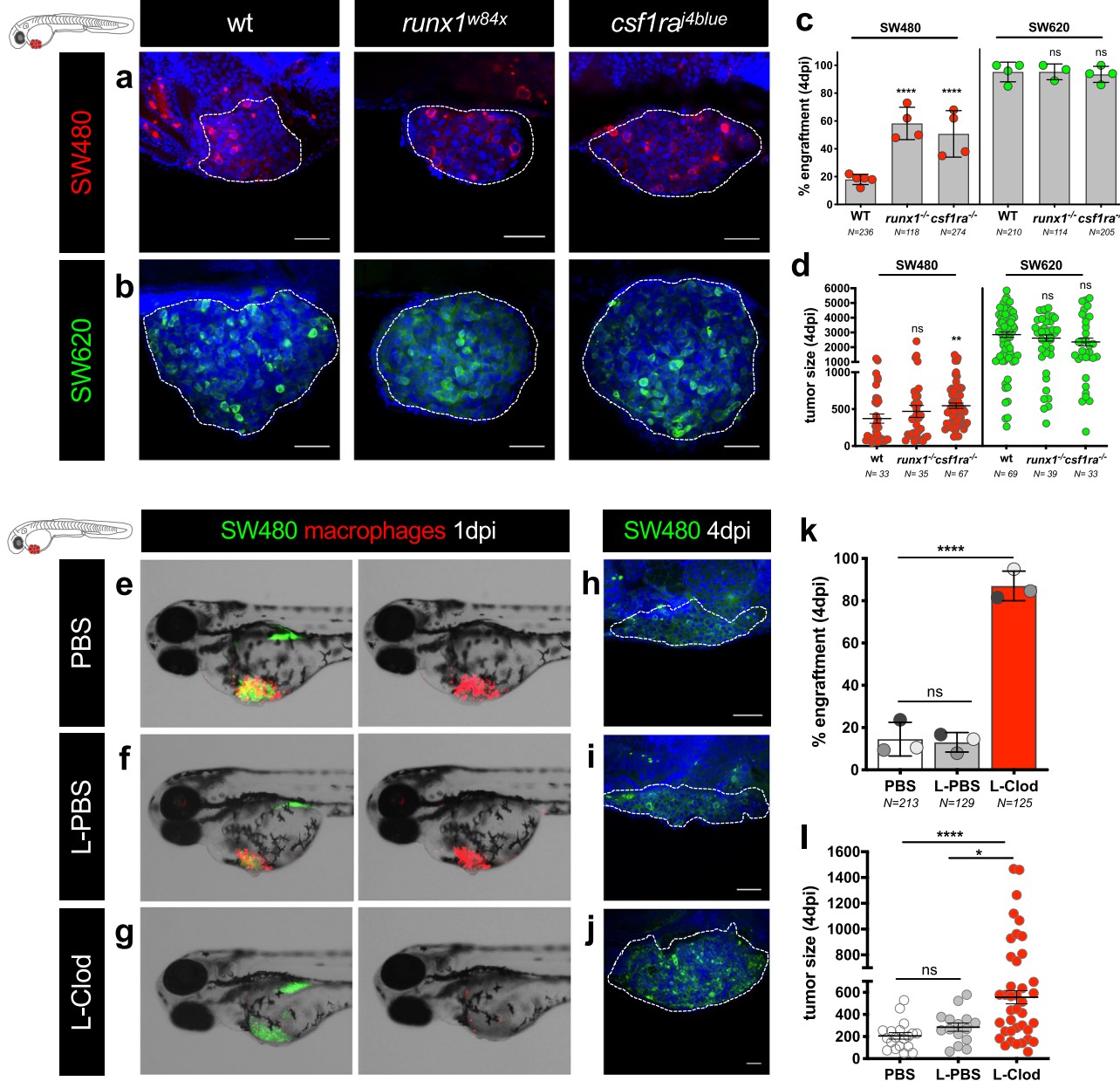

**Fig. 5 Zebrafish innate immune cells regulate clearance of SW480 tumor cells. a, b** Representative confocal images of SW480 and SW620 xenografts in *runx1^{w84x}* and *csf1rai^{i4blue}* (*panther*) mutants. SW480 were labeled in red and SW620 in green. **c** Quantification of engraftment in *runx1^{w84x}* and *csf1rai^{i4blue}* (*panther*) mutants and respective controls (Fisher exact test, ****P < 0.0001, SW620 wt vs SW620 *runx1^{w84x}* ns = 0.62, SW620 wt vs SW620 *panther* ns = 0.09). Error bars represent mean ± S.D. Each dot represents one independent experiment. **d** Quantification of tumor size in *runx1^{w84x}* and *csf1rai^{i4blue}* (*panther*) mutants and respective controls (unpaired two-sided Mann–Whitney test—SW480 wt vs SW480 *runx1^{w84x}* ns = 0.22, **P = 0.0013, SW620 wt vs SW620 *runx1^{w84x}* ns = 0.44, SW620 wt vs SW620 *panther* ns = 0.18). Error bars represent mean ± SEM, each dot represents one xenograft from 3 independent experiments. **e–j** Zebrafish embryos with 2 dpf were injected simultaneously with SW480 tumor cells (in green) with PBS (control), with L-PBS or with L-Clodronate liposomes into *Tg(mpeg1:mcherry)* background (macrophages in red). **e–g** Representative fluorescence stereoscope images of SW480 xenografts at 1 dpi in the different conditions. **h–j** Representative confocal images of SW480 xenografts at 4 dpi. **k** Quantification of engraftment: Fisher exact test ns = 0.83, ****P < 0.0001; error bars represent mean ± S.D.; each dot represents one independent experiment, and each set of independent experiment is represented in a different gray color. **l** Quantification of tumor size—no. of tumor cells (unpaired two-sided Mann–Whitney test ns = 0.062, ****P < 0.0001, *P = 0.022, error bars represent mean ± SEM) in the different experimental conditions at 4 dpi, each dot represents one xenograft from 3 independent experiments. Scale bars: 50 μm. White dashed lines encircle tumor areas. Nuclei are stained with DAPI. *N* is depicted in the chart. See also Supplementary Fig. 6. Source data are provided as a Source data file.

macrophage population than SW620 or in MIX (Fig. 6a–c, d, P = 0.016, see Supplementary Fig. 7 for gating strategy). Also, like our previous zebrafish results, SW620 cells became the dominant clone in MIX mouse xenografts (Fig. 6e, *P = 0.029).

To test whether mouse macrophages can actively modulate SW480 tumors, macrophages were depleted with L-clodronate. Results show that similarly to zebrafish, macrophage depletion leads to an increase in tumor size (Fig. 6f, **P = 0.007,

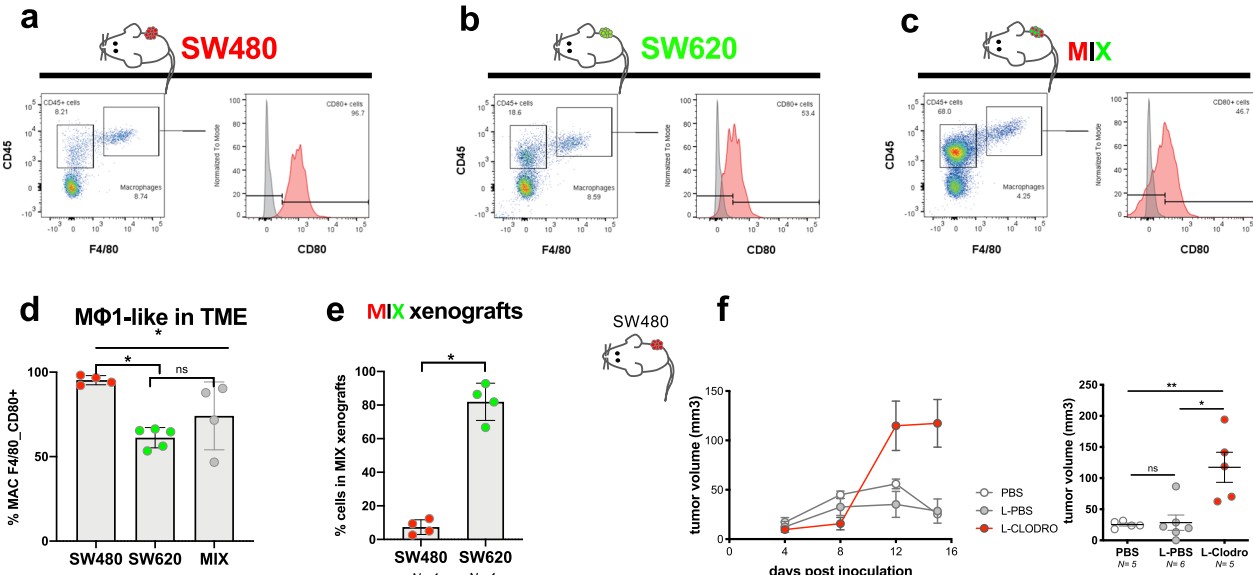

**Fig. 6 Mice xenografts display similar TME behavior as zebrafish. a–c** Representative graphs of flow cytometry analysis of the TME of SW480, SW620, and MIX mouse ($Rag1^{-/-}$ C57BL/6J) xenografts at 3 weeks post inoculation. **d** Quantification of double positive F4/80, CD80 macrophage in each TME, SW620 vs SW480. *$P$ = 0.016, Cohen's D $g$ = 6.2; SW620 vs MIX ns = 0.29, Cohen's D $g$ = 0.83; SW480 vs MIX *$P$ = 0.029, Cohen's D $g$ = 1.28). **e** Quantification of the percentage of each clone in MIX mice xenografts, *$P$ = 0.029, Cohen's D $g$ = 7.68. **d, e** Data from quantification of flow cytometry analysis. Error bars represent mean ± S.D. **f** Growth curves of SW480 tumors treated with PBS, L-PBS, or L-Clodronate mice (**f** ns = 0.43, Cohen's D $g$ = 0.12, **P = 0.007, Cohen's D $g$ = 2.16, *$P$ = 0.017, Cohen's D $g$ = 1.92). Error bars represent mean ± SEM. All data were analyzed using unpaired two-sided Mann–Whitney test. To avoid macrophage repopulation, mice were injected every 4 days (see "Methods"). $N$ is depicted in the chart. Each dot represents one mouse xenograft. Source data are provided as a Source data file.

*$P$ = 0.017), suggesting that the role of macrophages in SW480 TME is conserved across species.

As a comment on the differences of the models, the analysis of the murine model was performed ~24 days post tumor injection, while our zebrafish TME analysis was performed at 1 and 4 dpi, i.e., a discrepancy of ~20 days. We believe that the zebrafish model allows for an immediate snapshot of the "tumor state", but the murine model allows to study how these tumor-TME interactions evolve along time. Therefore, this "timing" issue can account for some differences, and does not necessarily undermine one model or the other.

**Innate immunoediting in zebrafish xenografts**. Next, we aimed at analyzing if engrafted zebrafish SW480 tumors were undergoing innate immunoediting, and therefore, would be able to escape host innate immunity.

To this end, seven SW480 tumors were dissected at 4 dpi, from an experiment that yielded ~12% engraftment. Dissected tumors were then expanded in vitro for three passages (Fig. 7a) and these (SW480zEscapers cells) were next injected into 2 dpf zebrafish embryos. Engraftment, tumor size, proliferation, apoptosis, and macrophage infiltration were quantified and compared to parental cells. Strikingly, SW480zEscapers engrafted much more efficiently (from an average of ~20% in parental to ~60% in SW480zEscapers, $P$ < 0.0001) (Fig. 7b, c) and tumor size increased in relation to parental tumors (Fig. 7d, $P$ < 0.0001). Interestingly, we could not detect a higher proliferation rate in these tumors (Fig. 7e) and apoptosis levels were slightly increased (Fig. 7f). Thus, these results reinforce the idea that proliferation and apoptosis are not the main drivers of engraftment/clearance. Importantly, the macrophage infiltrate was significantly reduced in these tumors (Fig. 7g, <0.0001). These results suggest that innate immunity plays a critical role in immunoediting cancer cells toward tumorigenesis.

**Clearance and expansion of different SW480 subclones**. To investigate the molecular alterations that might underlie the emergence of SW480 escapers (as well as the subclones that get cleared), we performed single-cell transcriptomic profiling. We injected SW480 parental cells (GFP transfected) and then dissected tumors at 2 time-points for single-cell RNA-seq (scRNA-seq): 1 dpi (where all subclones should be present) and at 4 dpi (where only the subclones that escape clearance are present) (Fig. 8a). Dissociated single cells were sorted by fluorescence-activated cell sorting (FACS) into 384-well plates for scRNAseq SORT-seq[35]: 3 plates for the first timepoint and 2 plates for the second (Fig. 8a, b and see Supplementary Fig. 8a for quality control). Cells were pooled and clustered according to their gene expression profiles using Seurat[36], resulting in six different cell clusters (cell states), which were visualized using the uniform manifold approximation and projection (UMAP) approach[37] (Fig. 8b, c, see Supplementary Fig. 8 for PCA and heatmap showing the differential gene expression between different clusters).

Comparing the clusters' frequency between 1 and 4 days, it was possible to follow how the various tumor clusters changed (Fig. 8) but also the dynamics of the signaling pathways (Supplementary Fig. 8d). Interestingly, two cell clusters (1 and 4) almost disappear in just 3 days, whereas others maintain their frequency (0 and 2) but others clearly expand (3 and 5). These results suggest that some clusters were cleared (1 and 4), while others were able to evade innate immune detection and were therefore maintained (0, 2, 3, and 5) (Fig. 8e).

In cluster 1, whose frequency was strikingly reduced, enrichment pathway analysis showed the activation of innate immune-related pathways as the interferon pathway (Myd88 independent TLR cascade and DNA-dependent activation of IFN-regulatory factors) as well as several inflammatory cytokines (e.g., CX3CL1, CXCL1) (Fig. 8d, e Supplementary Fig. 9a-c, Supplementary Data 2). These cytokines are known to act as chemoattractants for

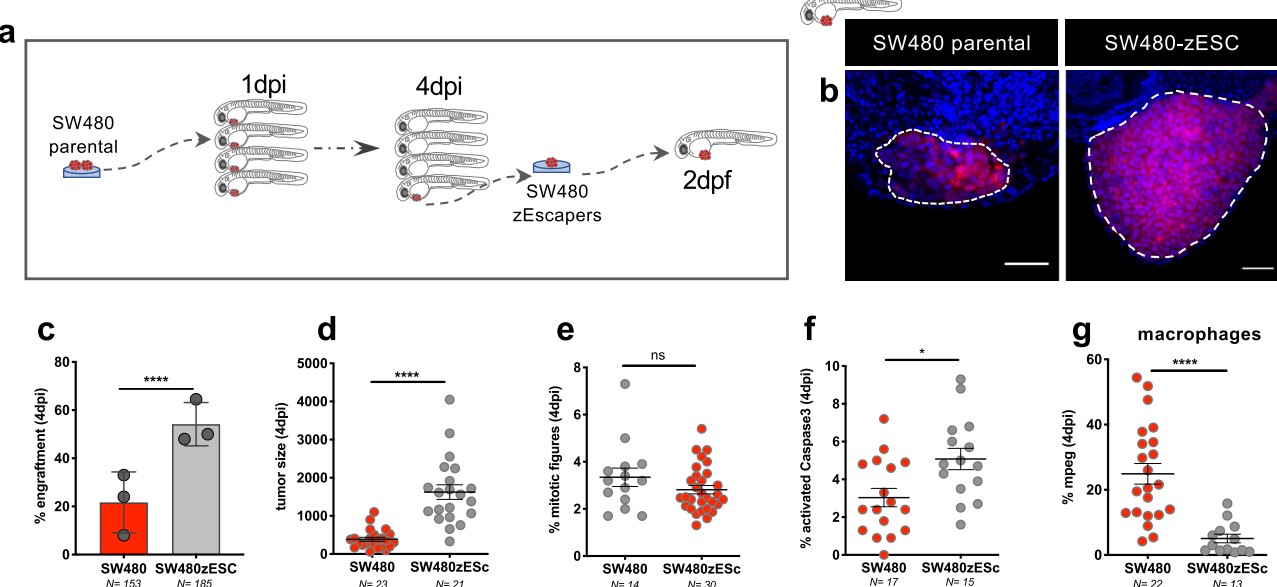

**Fig. 7 Innate immunoediting in zebrafish xenografts. a** Schematic illustration of SW480 escaper cells selection from SW480 parental xenografted (see "Methods" for more info). **b** Representative confocal images of tumoral masses of SW480 parental and SW480zEscapers xenografts at 4 dpi. **c** Quantification of engraftment at 4 dpi (Fisher exact test ****$P < 0.0001$). Error bars represent mean ± S.D. Each dot represents one independent experiment. **d** Quantification of tumor size—no. of tumor cells, at 4 dpi (unpaired two-sided Mann–Whitney test ****$P < 0.0001$). **e** Quantification of mitotic tumor cells at 4 dpi (unpaired two-sided Mann–Whitney test ns = 0.25). **f** Quantification of apoptotic tumor cells at 4 dpi (unpaired two-sided Mann–Whitney test *$P = 0.01$). **g** Quantification of macrophage present in the TME of SW480 parental versus SW480Zesc at 4 dpi (unpaired two-sided Mann–Whitney test ****$P < 0.0001$). In dot plots, error bars represent mean ± SEM. Scale bars: 50 μm. Dashed lines encircle tumor areas. Nuclei are stained with DAPI. N is depicted in the chart. Each dot represents one xenograft. Data of SW480zEscapers results from three independent injections. Source data are provided as a Source data file.

various immune cells; the large CX3CL1/fractalkine attracting T cells and monocytes[38], whereas the small chemokine CXCL1 acts in particular to attract neutrophils during inflammation[39]. Their increased expression in subclones that decrease frequency in the tumor might contribute to this clearance.

In contrast, an enrichment of IL10 immunosuppressive related signaling was observed in cluster 3 (which is expanded at 4 dpi), suggesting that IL10 signaling might protect SW480_zEscapers from clearance (Fig. 8d, e and Supplementary Fig. 9d).

Comparison analysis between expanding cluster 3 vs cleared clusters (1 and 4) reveals the opposing dynamics between IL10 signaling and IFNγ (Supplementary Fig. 9e).

SW480 and SW620 cells have been previously ascribed to a stem-like subtype, with high expression of Wnt signaling targets as well as other stem cell and mesenchymal genes, together with low expression of differentiation markers[40]. As expected, we could identify enrichment of Wnt and Notch pathways (Supplementary Fig. 10), as these are major players in the maintenance of the stem cell state and the regulation of differentiation of transit-amplifying (TA) progenitors[41].

Wnt signaling seemed to be highly active in cluster 3, as highlighted by the higher expression of various pathway components (NOTUM, APCDD1, and AXIN2, see Supplementary Fig. 10a and Fig. 8e). In contrast, Notch activation was uniquely predominant in cluster 1, as evidenced by the high expression of HES1, HES5, and HEY2/L genes (Supplementary Fig. 10b), which are canonical downstream targets and effectors of the pathway[42]. Notch signaling, besides contributing to the stem cell state, is essential in the decision between absorptive TA progenitors (NOTCH_ON) vs secretory TA progenitors (NOTCH_OFF)[41]. Since NOTCH-ON cluster 1 was mostly cleared, we wondered if the other expanding clones had markers for the "opposing" secretory-like fate. Indeed, we observed that expression of ASCL-2,

TFF3, and PROX1[43,44] were highly enriched in cluster 3 (Supplementary Fig. 10c, d). Interestingly, the expanded cluster 5 seemed to have an enrichment in Tert and Dll4, suggesting that this cluster may represent the quiescent-like progenitor pool known as +4[45] (Fig. 8e, Supplementary Fig. 10c, d).

In summary, our results show the clearance of specific regressors' subclones expressing IFN related signaling and Notch activation, as well as the expansion of subclones that express IL10 suppressive pathway with expansion of Wnt and secretory-like "states" (cluster 3), as well as a putative "quiescent"-like progenitor state (cluster 5) (Fig. 8e).

## Discussion

In the present study, we take advantage of the fast zebrafish larvae xenograft model to study the crosstalk between human cancer cells and the innate immune system. Previous work[15] suggested that although most human tumors engraft well, some are cleared from the zebrafish host. Here, we studied a pair of human CRC cells derived from the same patient at different stages of tumor progression—SW480 from the primary tumor, and SW620 from a lymph node metastasis isolated 6 months later. While SW480_regressors engraft poorly and are mostly cleared in 4 days, SW620_progressors have high engraftment rates. Gene expression assessed by RNA-seq of these tumors revealed the involvement of innate immune-related pathways that may contribute to this phenotype. Indeed, we found that SW480 cells recruit neutrophils and macrophages more efficiently than SW620. However, SW620 can polarize macrophages toward a M2-like pro-tumoral phenotype. Genetic and chemical depletion of myeloid cells demonstrate that macrophages and neutrophils play a crucial role in tumor clearance. We also performed re-transplantation experiments of in vivo selected tumors and

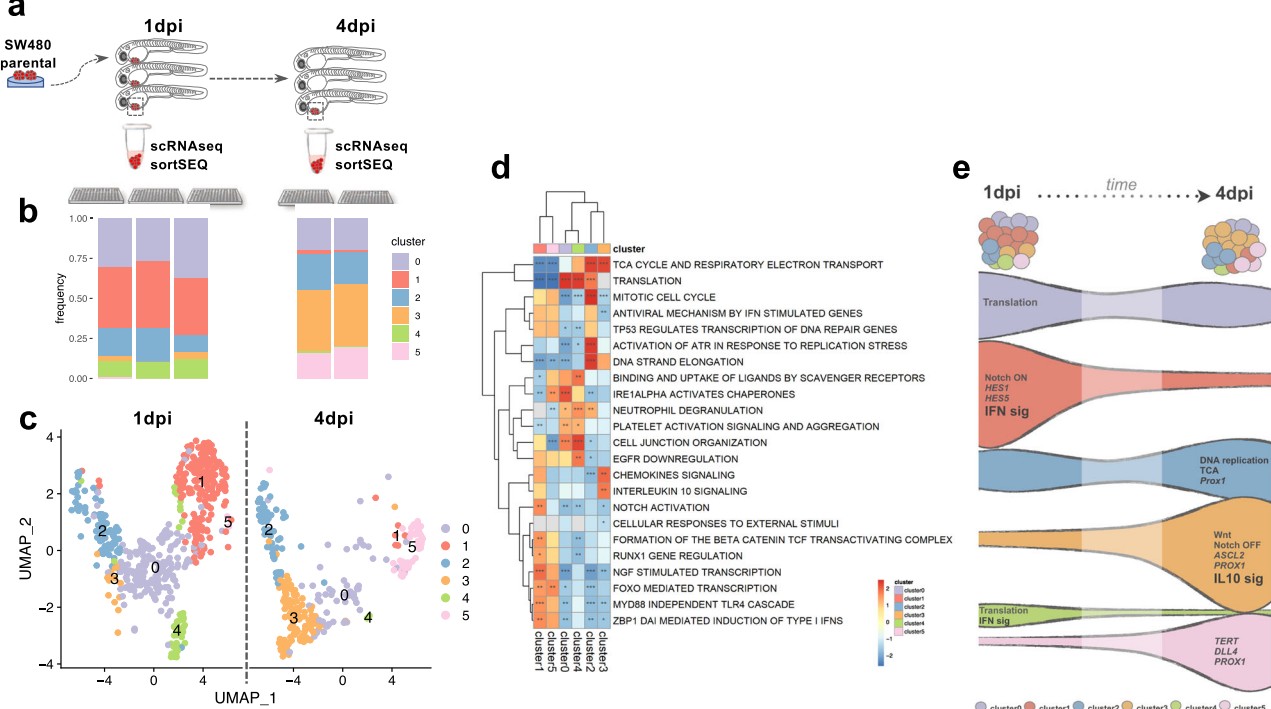

**Fig. 8 Single-cell transcriptome profiling reveals the clearance and expansion of different SW480 subclones. a** Schematic illustration of the design of the experiment. SW480 cells were injected into 2 dpf zebrafish embryos, and at 1 and 4 dpi, tumors were dissected and processed for scRNAseq. **b** Relative frequencies of the cell clusters present in each library replicate. **c** Uniform Manifold Approximation and Projection (UMAP), representing the relative similarity between individual cells, colored by cell cluster and divided by timepoints 1 and 4 dpi. **d** Heatmap representation of normalized enrichment scores (NES) of representative pathways with statistically significant (adjusted *P*-value < 0.05) enrichment in gene set enrichment analysis (GSEA), comparing the gene expression of each cellular subgroup to all the others. Red colors mean that genes in that pathway tend to be more expressed in that cellular subcluster, while blue means that genes tend to be less expressed. Significant NES values are marked with asterisk (Fisher exact test *: adjusted *P*-value < 0.05; **: adjusted *P*-value < 0.01; ***: adjusted *P*-value < 0.001). Gray colors are cases where a NES value could not be obtained and should be considered non-significant (see Supplementary Data 2 for GSE values). **e** Schematic illustration of expansion/reduction of each cluster from 1 to 4 dpi with the most representative pathways and genes.

showed that these tumors engrafted more efficiently and generated bigger tumors with reduced macrophage infiltrates. These results suggest that zebrafish innate immunity can immune-shape tumors toward tumorigenesis.

Finally, single-cell transcriptome analysis clearly shows a fast tumor selection process, with clearance and expansion of specific subclones or "cell states" in just 3 days. In accordance with our hypothesis, the "cleared"-regressor subclones were associated with activation of immune-inflammatory pathways and escaper-expanded subclones with an enrichment of IL10 immune suppressor pathway and a secretory-like fate. Interestingly, we observed a clearance of subclones with active Notch signaling. The role of Notch signaling and of the identified immune-related pathways will be the subject of future investigation.

The concept of immunoediting has been mainly focused on adaptive immunity[46,47], and only one study, to our knowledge[16], has shown that innate immunity on its own is able to perform immunoediting. O'Sullivan describes the role of NK cells in educating macrophages toward an anti-tumoral state that act as crucial effectors in immunoediting in a RAG2−/− x γc (−/−) mouse model of induced sarcoma. In the present study, we show that not only macrophages play an essential role but also that neutrophils contribute to clearance and therefore to immunoediting. We may speculate that, as NK cells educate macrophages in the O'Sullivan model, neutrophils might also interact with macrophages, possibly "re-educating" them toward an anti-tumoral phenotype.

These results, which show that neutrophils and macrophages may have an active anti-tumoral role, are in accordance with CRC clinical data that suggest that TAMs and TANs are associated with a favorable prognosis, especially in early stages[48–52]. On the other hand, in other types of cancer such as lung, gastric, gynecological, and breast cancer, high infiltration of TAMs correlates with a poor clinical prognosis[6,53].

Although the presence of macrophages might indicate a poor prognosis, macrophages can be in an anti-tumoral state and therefore tumoricidal, becoming instead a good prognosis. Or on the other hand, if the tumor has been able to communicate with the nearby macrophages and "talked them" into becoming pro-tumoral and immunosuppressive then it will be indeed an indicative of a bad prognosis. Thus, identifying the functional state of the TME cells becomes fundamental to anticipate prognosis and also response to immunotherapy, i.e., defining a hot (permissive) or cold (immunosuppressed) TME. Many different markers have emerged to classify immune cell types as pro- or anti-tumoral. However, no universal robust marker which may help guide treatment decisions, has been found so far. Most studies use a battery of different molecules to identify the macrophage subtypes[52–57], although this battery also varies between studies. This is probably due to the amazing plasticity and diversity of the different macrophage populations. There are not two static states but multiple states that are dynamic and interchangeable[58–60]. Also, there are numerous types of macrophages with different embryonic origins, leading to a huge heterogeneity

of phenotypes and functions. Consequently, to find a good and universal marker of the innate functional status has been a major hurdle, that has not been yet achieved[59].

Numerous studies have shown that the zebrafish model can respond to human tumor angiogenic cues and therefore be used as a reporter of the angiogenic potential of tumor cells[61,62]. Here, we show that zebrafish can also "read" the innate immune cues and reconstitute an innate microenvironment in just 4 days. We propose that by analyzing the engraftment/clearance in wild-type (wt) and mutants as well as using reporters, such as TNFa, it is possible to infer the function of these innate immune cells.

Our results are opening the possibility of using the zebrafish Avatar model as a living biomarker to infer the innate TME state, i.e., reveal an anti-tumoral state (immune permissive/hot) or pro-tumoral (and immune suppressive/cold). Importantly, this could have a prognostic value and possibly help select patients that can benefit from TME-based therapies, such as immunotherapy. In addition, we also propose that future experiments can use zebrafish xenografts to study innate immune suppressive mechanisms and possibly find new therapeutics to enhance immunotherapy.

## Methods

**Zebrafish welfare and handling**. Zebrafish (*Danio rerio*) model was handled and maintained according to the standard protocols of the European Animal Welfare Legislation, Directive 2010/63/EU (European Commission, 2016) and Champalimaud Fish Platform. All protocols were approved by the Champalimaud Animal Ethical Committee and Portuguese institutional organizations—ORBEA (Órgão de Bem-Estar e Ética Animal/Animal Welfare and Ethics Body) and DGAV (Direção Geral de Alimentação e Veterinária/Directorate General for Food and Veterinary).

**Zebrafish transgenic and mutant lines**. According to the purpose of each experiment, different genetically modified zebrafish lines were used in this study: *Tg(mpx:eGFP)*[23], *Tg(mpeg1:mCherry-F)*[24], *Tg(mpeg1:mCherry-F; tnfa:GFP-F)*[24], *runx1*[w84x] mutant[26], and *csf1ra*[j4blue] (*panther*) mutant[27]. Wild-type Tubingen strain or Casper mutants were used as control and as a background line for the experiments.

**Human tissue processing**. Human samples used for zebrafish patient-derived xenograft (zAvatars) establishment were obtained from Champalimaud Hospital and Prof Fernando Fonseca Hospital with written informed consent. The study was approved by both Hospital Ethics Committees.

Neoplastic colorectal tissues were obtained from surgically resected specimens. Human tissue processing protocol was performed as previously described[15]. In brief, samples were washed in ice-cold 1X-PBS, chopped into small pieces, and cryopreserved in 90% FBS 10% DMSO. Cryopreserved human primary tumor tissue was defrosted, further washed, and minced in mix1 (DMEM-F12 (Gibco), 60%FBS (Sigma), Y-27632 10 μM (Cliniscience), Primocin 100 μg/ml (Invivogen), Putrescin 10 μg/ml (Sigma-Aldrich), Nicotinamide 10 mM (Sigma-Aldrich), and digested with Liberase (Roche) for 5–10 min at 37 °C. Tumor cell suspension was passed through a 70 μm cell strainer and centrifuged at 250 × g for 4 min at 4 °C. For cell labeling, tumor cells were incubated with CM-DiI (1:100) in mix1 but without FBS and supplemented with DNase I 5 U/ml (Fermentas), for 15 min at 37 °C. Cells were resuspended in mix1 supplemented with human EGF (50 ng·mL⁻¹, Peprotech) at final concentration of ~0.25 × 10⁶ cells per milliliter.

**Human cancer cell lines and culture**. Human breast cancer cell lines Hs578T, MDA-MB-231, and MDA-MB-468 were kindly provided by Mónica Bettencourt Dias' Lab (Instituto Gulbenkian da Ciência).

Human colorectal cancer cell lines SW480, SW620, and HT29 were purchased from American Type Culture Collection (ATCC), whereas HCT116 and Hke3 isogenic cell lines were kindly provided by Ângela Relógio (Charité Medical University of Berlin). SW48 cell line was provided by Luis Costa Lab (ATCC, Instituto de Medicina Molecular). All cell lines were kept and grown in Dulbecco's modified Eagle medium (DMEM) High Glucose (Biowest) and supplemented with 10% fetal bovine serum (FBS) (Sigma-Aldrich) and antibiotics (100 U ml⁻¹ penicillin and 100 μg ml⁻¹ streptomycin, Hyclone) in a humidified 5% CO₂ atmosphere at 37 °C. All cell lines were authenticated through short tandem repeat (STR) profile analysis and tested routinely for mycoplasma contamination.

**Cell staining**. Tumor cells were grown to 70% confluence, washed with Dulbecco's phosphate-buffered saline (DPBS) 1X (Biowest) and stained in a flask with lipophilic dyes—Vybrant CM-DiI (4 μl/ml in DPBS 1X), green CMFDA (1 μl/ml in DPBS 1X, 1 mM stock), or Deep Red Cell Tracker (1 μl/ml in DPBS 1X, 10 mM

stock) (Life Technologies), for 10 min at 37 °C, in darkness. Cells were washed with DPBS and detached with 2 mM EDTA by scrapping. Cell suspension was collected to 1.5 ml eppendorfs, centrifuged at 250 × g, for 4 min at 4 °C, and resuspended in DMEM. Cell viability was assessed by trypan blue exclusion method, and cell number was determined by hemocytometer counting. Cells were resuspended in DPBS 1X to a final concentration of 0.25 × 10⁶ cells/μl.

**Zebrafish xenografts**. Fluorescently labeled cancer cells were injected using borosilicate glass microcapillaries under a fluorescence scope (Zeiss Axio Zoom. V16) with a mechanical micropipetor attached (World Precision Instruments, Pneumatic Pico pump PV820). Approximately 500–1000 cells were injected into the periviteline space (PVS) of 2 dpf zebrafish embryo, previously anesthetized with Tricaine 1X (Sigma-Aldrich). After injection, zebrafish xenografts remained for ~10 min in Tricaine 1X and then transferred to E3 medium and kept at 34 °C. At 1 dpi, zebrafish xenografts were screened according to the presence or absence of tumoral mass. Xenografts with cells in the yolk sac or cellular debris were discarded, whereas successfully ones were grouped according to their tumor size, which was classified by comparison with eye's size. Every day xenografts were checked—dead ones removed and E3 medium refreshed. Four days after injection the engraftment rate was calculated (formula below) and zebrafish xenografts were sacrificed, fixed with 4% (v/v) Formaldehyde (FA) (Thermo Scientific) at 4 °C overnight and preserved at −20 °C in 100% (v/v) methanol.

Xenograft engraftment calculation

$$\text{Engraftment (\%)} = \frac{\text{no. of xenografts with tumor at 4 dpi}}{\text{total no. of xenografts at 4 dpi}} \times 100$$

**Ratios of SW480 and SW620**. SW480 and SW620, prepared for injection as mentioned above, were mixed in different proportions right before injection to generate the following ratios:

Mix 3:1—75% SW480 + 25% SW620
Mix 1:1—50% SW480 + 50% SW620
Mix 1:3—25% SW480 + 75% SW620

**Chemo/radio treatment of zebrafish xenografts and zAvatars**. At 1 dpi, zebrafish xenografts with the same tumor size were randomly distributed in the treatment groups: control E3 medium and FOLFOX in E3 (4.2 mM 5-FU, 0.18 mM folinic acid, 0.08 mM oxaliplatin) for three consecutive days, replaced daily. Maximum tolerated concentration in zebrafish larvae was determined as previously described[15]. Single dose of 25 Gy was delivered to zAvatars at 1 dpi as described in ref. [63] and 3 days after the experiment ended. In brief, Irradiation procedures and regimens were adapted for zebrafish xenografts by the Champalimaud Foundation Radiation Oncology Department. The 6MV X-rays beams with 25 Gy were calculated with the same algorithm used in clinical practice (ECLIPSE, Varian Medical System, CA) and was delivered via a linear accelerator (Truebeam, Varian Medical Systems, CA). Irradiation was targeted to the center of a defined area of 30 × 30 cm from a 6-well plate with the anesthetized zebrafish (6 mL of E3 medium per well, ~12 xenografts per well). The well plates were positioned with a source-to-surface distance of 100 cm.

**Zebrafish macrophage ablation with clodronate liposomes**. For the selective depletion of macrophages, Liposomes-encapsulated PBS (L-PBS) and Liposomes-encapsulated clodronate (L-Clodronate) were purchased from Liposoma. At the time of cell resuspension immediately prior to cell microinjection into zebrafish, cells were resuspended either in PBS, L-PBS, or L-Clodronate at a final concentration of 0.25 × 10⁶ cells/μL.

**Imaging and analysis of zebrafish xenografts**. All images were obtained using a Zeiss LSM 710 fluorescence confocal microscope, generally with a 5 μm interval in a total of ~60 μm stack using the z-stack function. Generated images were processed using the FIJI/ImageJ software. Some of the acquired z-stacks were projected using maximum intensity projection. Number of cells was quantified with ImageJ software Cell counter plugin.

To assess tumor size, three representative slices of the tumor, from the top (Zfirst), middle (Zmidle), and bottom (Zlast), per z-stack per xenograft were analyzed and a proxy of total cell number of the entire tumor (DAPI nuclei) was estimated as follows:

$$\text{tumor size} = \frac{\text{no. of DAPI cells Zfirst + no. of DAPI cells Zmidle + no. of DAPI cells Zlast}}{\text{total number of slices} \times 1.5}$$

The 1.5 correction number was estimated to these CRC cells that have a nuclei with an average of 10–12 μm of diameter. Number of mitotic figures, activated caspase-3, macrophages, neutrophils, M1 and M2-like TNFa+ and TNFa− macrophages, as well as other inflammatory cells were counted in every slice, starting in the first and finishing in the last slice of the tumor. To get the percentage of each, raw number was divided by tumor size.

**Bulk RNA-seq sample preparation**. SW480 and SW620 tumors were dissected from zebrafish xenografts at 2 dpi. A pool of ~30 tumors from each type of tumor

and independent experiments was collected in RNA*later* solution (#AM7020, Ambion) and kept at −20 °C until RNA extraction. The engraftment rate (determined at 4 dpi with remaining zebrafish xenografts) of SW480 and SW620 used for gene expression analysis was the following: SW480_B—8%, SW480_A—30%, and SW480_7—31.5%; SW620_1—92.4%, SW620_2—92%, SW620_5—97.2%, and SW620_7—98.3%. To study the genetic signatures of the underlying observed phenotypes (regressors and progressors), total RNA was extracted from the dissected tumors using Trizol reagent (Invitrogen Life Technologies, Carlsbad, CA, USA) and further purified with RNeasy Plus Micro Kit (Qiagen), in accordance with the manufacturer's instructions.

**RNA-seq analysis.** mRNA-libraries were prepared using the Smart-seq2 protocol (Illumina, USA). Samples were sequenced by Next-Seq 500 Illumina sequencer and unstranded single-end mRNA-seq libraries of 76 bp were obtained. An average of ~38 million reads per sample. These RNA-seq libraries contain a mixture of human and zebrafish RNA derived from the xenograft as well as the host cells infiltrating it. After quality control assessment with FastQC[64] (v0.11.7) and low quality reads filtering with Trimmomatic[65] (v0.38), all sequenced libraries were quantified with Salmon[66] (v0.13.1 using the respective transcript human annotations (Hg38) from the Ensembl genome database project). For downstream analysis package Tximport[67] was used, to import transcript lengths and abundance estimates and export (estimated) count matrices. And differential expression analysis was performed using Limma[68]. Genes with a FDR < 0.05 and absolute log2 foldchange >1 were considered significant.

**Pathway enrichment analysis of a ranked gene list using GSEA.** Pathway enrichment analysis helps us gain biological insight into large gene lists typically resulting from high throughput experiments. It identifies biological pathways that are enriched in the gene list more than expected by chance. A ranked gene list obtained from SW480 low engraftment versus SW620 high engraftment differential expression analysis was input to GSEA PreRank[19] (v4.0.2, Broad Institute, Cambridge, MA) as RNK file. We used curated gene sets from Molecular Signatures Database[69] (v7.0, Hallmarks and Canonical Pathways including KEGG and REACTOME). We then ran GSEA PreRank using the default weighted statistic. The thresholds for significance were determined by permutation analysis (1000 permutations), selecting the enriched pathways with a false discovery rate (FDR) < 0.07.

**Single-cell RNA-seq preparation.** SW480 tumors were dissected from zebrafish xenografts at 1 and 4 dpi. A pool of ~100 tumors was collected to 1 ml of DMEM High Glucose (Biowest), 10 μM of Anoikis inhibitor Y-27632 2HCl (#S1049, Selleckchem) and 5U of Dnase (#EN0521, Thermofisher). Tumors were then digested by adding 5 μl of liberase (5 mg/ml, #05401020001, Roche), for 2–3′ at 37 °C and centrifuged at 300 × g, for 4′ at 4 °C. Cell suspension was resuspended in DPBS 1X (Biowest), EDTA 2 mM (Sigma), FBS 2% (Sigma) and Hepes (Fisher Scientific) 25 mM, filtered with 70 μm strainer, and DAPI (5 μg/ml) was added as a control for live-cell selection in FACS (stained DAPI cells were dead and DAPI negative_live were sorted). Cells were kept on ice and FACS sorted into 384-well plates containing 384 primers and Mineral oil (Sigma).

**Single-cell RNA-seq analysis.** During sequencing, Read 1 was assigned 26 base pairs and was used for identification of the Illumina library barcode, cell barcode and Unique Molecular Identifiers (UMI). R2 was assigned 60 base pairs and used to map the Human reference transcriptome with BWA[70]. Unique barcode gene counts were used for further processing in Seurat[36]. Only genes present in at least 10 cells were considered. Moreover, we only considered barcodes with counts in more than 2000 genes and with <25% of counts in mitochondrial genes. After quality control and filtering, we were left with a total of 533 human GFP positive cells from the first timepoint and 293 human GFP positive cells from the second. After normalization, the 2000 most variable genes were used for dimensionality reduction and clustering. We chose clustering parameters empirically to provide a balance between the number of clusters and their size. We then ran GSEA analysis for each cluster by ranking genes according to their differences in gene expression to the other clusters. Cells were displayed in a 2-dimensional plot using uniform manifold approximation and projection (UMAP)[37]. Normalized expression values of genes from selected enriched pathways were also displayed as violin plots or heatmaps. The Normalized expression values of genes used for visualization correspond to the log2 of total cell counts divided by 10,000 (very similar to the traditional CPM). For heatmaps, expression values were displayed scaled by gene (row z-score).

**Mouse welfare and strains.** Mice experiments and corresponding protocols were approved by the Champalimaud Animal Ethical Committee and portuguese institutional organizations—ORBEA (Órgão de Bem-Estar e Ética Animal/Animal Welfare and Ethics Body) and DGAV (Direção Geral de Alimentação e Veterinária/Directorate General for Food and Veterinary).

Rag1−/− C57BL/6J mice were bred at 23 °C, with 40–60% relative humidity, 12 h light cycle (8 am–8 pm) by the animal facility of Champalimaud Vivarium, Lisbon, Portugal.

**SW480 and SW620 transduction by lentiviral infection.** To facilitate posterior quantification of each population on mice experiments, we first generated SW480 and SW620 cell expressing GFP and tomato fluorescent proteins, respectively (false colors in the figures).

SW480 and SW620 cells were seeded (1 × 10⁶ cells per well) in 6-well plates and incubated at 37 °C ON. In the following day, a range of dilutions (1:50 up to 1:1000) of lentivirus vectors were added to each well. DMEM supplemented with 10%FBS, 1%P/S, and 8 μg/mL polybrene was used to enhance transduction efficiency. Twenty-four hours later, medium was replaced to obtain stable transduced cells and maintained at 37 °C. Untransduced cells with the same antibiotics were used as controls.

Cells were expanded and the transduction efficiency was measured by flow cytometry (BD LSRFortessa™ X-20 cell analyser—Biosciences). Cells were then sorted (BD FACSAria Fusion) using FACS Diva software v8, with a 99% of purity.

**Mouse SW480 xenografts and macrophage ablation.** Approximately, 1 × 10⁶ SW480 cells were resuspended with 1:1 matrigel (reduced growth factors, Corning) and one of the following conditions: PBS, L-PBS, or L-Clodronate. SW480 were injected subcutaneously in the right flank of 8-week-old Rag1−/− C57BL/6J mice (N = 5 per group).

For macrophage depletion, L-Clodronate was administrated right upon tumor cell inoculation through intravenous injection (retro-orbital injection). The same protocol was performed for PBS or L-PBS controls. To avoid macrophage repopulation, the same treatment conditions were injected every 4 days. Tumor size was measured once a week using caliper measurements and tumor volumes were calculated according to a standard formula:

$$\frac{4}{3}\pi \times (\text{Short axis of the tumor}/2)^2 \times (\text{Long axis of the tumor}/2)$$

At the end of the experiment, mice were euthanized with carbon dioxide, and tumor size was measured and immediately fixed in 4% FA.

**Mouse SW480, SW620, or MIX 1:1 xenografts.** Approximately, 1 × 10⁶ tumor cells (SW480, SW620, or MIX 1:1) were resuspended with 1:1 matrigel (reduced growth factors, Corning) and subcutaneously injected in the right flank of 7–10-week-old Rag1−/− C57BL/6J mice.

Tumor size was measured every 3–4 days using caliper measurements and tumor volumes were calculated according to a standard formula of:

$$\frac{4}{3}\pi \times (\text{Short axis of the tumor}/2)^2 \times (\text{Long axis of the tumor}/2)$$

At the end of the experiment, mice were euthanized with carbon dioxide and tumor size was measured and immediately fixed in 4% (v/v) FA.

**Mouse xenografts tumor isolation and staining.** Tumor-bearing mice were euthanized according to approved guidelines with carbon dioxide three weeks (~24 days) after inoculation of cancer cells. Subcutaneous tumors were resected and measured with a caliper. Tumors were thoroughly minced with scalpels, transferred to 1.5 mL Eppendorf tubes, and digested in 1 ml of PBS 1X containing 10 μl of Liberase TM (5 mg/ml) (Sigma) and 3 μl of DNAse I (Thermo Scientific) for 30 min at 37 °C. Digested suspension was filtered through a 40 μm mesh into a 15 mL Falcon tube. Digestion was then blocked by addition of buffer containing 800 ml of HBSS (Corning), 2 ml of EDTA 0.5 M, and 0.1 g of BSA. Tubes were centrifuged 10 min at 300 × g. Pellet was resuspended in FACS buffer (PBS 1 × wo Ca/Mg, EDTA 2 mM, and FBS 2%). Total viable cell yield per volume was determined using Trypan Blue and an automated cell counter. Tumor single-cell suspension was then stained for FACS analysis.

**Flow cytometry antibodies.** The following monoclonal antibodies were used for flow cytometric analysis of tumors: Live/dead discrimination (LIVE/DEAD® Fixable Aqua Dead Cell Stain Kit, 1:500, Thermo Scientific, #L34957), anti-human EpCAM CD326 (9C4, Biolegend, 5:100), APC/Cyanine7 anti-mouse CD45.2 (104, Biolegend, 1:200), anti-mouse F4/80 PE-Cyanine7 (BM8, eBioscience, 1:200), FITC anti-mouse CD80 (16-10A1, Biolegend, 1:200).

**Flow cytometry gating strategy for mouse xenografts.** Data were acquired using the BD LSRFortessa™ X-20 cell analyzer (Biosciences) and analyzed using FlowJo™ v10.6.1 software. Populations were determined as follows: Human cells (LIVE/DEAD⁻EpCAM⁺), SW620 cells (LIVE/DEAD⁻EpCAM⁺PE⁺FITC⁻), SW480 cells (LIVE/DEAD⁻EpCAM⁺PE⁻FITC⁺), mouse cells (LIVE/DEAD⁻ EpCAM⁻), Macrophages (LIVE/DEAD⁻EpCAM⁻CD45⁺F4/80⁺), anti-tumoral M1-like macrophages (LIVE/DEAD⁻EpCAM⁻CD45⁺F4/80⁺CD80⁺). See Supplementary Fig. 7.

**Generation of SW480zEscapers.** Parental SW480 cells expressing GFP (red false color in Fig. 7) were injected into the PVS of 2 dpf zebrafish embryos. At 1 dpi, zebrafish xenografts were scored according to the tumor size and kept in E3 medium at 34 °C until day 4. Tumors that persisted until 4 dpi were dissected and expanded in vitro for three passages in multi-well plates—which we named

SW480zEscapers. SW480zEscapers were injected in new zebrafish embryos with 2 dpf and engraftment rate was quantified at 4 dpi.

**Immunofluorescence**. Whole-mount immunofluorescence was performed starting hydration through methanol series (75% > 50% > 25%). Next, xenografts were permeabilized with 0.1% (w/v) Triton in PBS and blocked with a mixture of PBS 1X, BSA, DMSO, Triton 1% (w/v), and goat serum, for 1 h at room temperature. The xenografts were then incubated with primary antibody Anti-Cleaved caspase-3 (Asp175) (rabbit, Cell Signaling, 1:100, #9661) or Mpx (rabbit, GeneTex, 1:50, #gtx128379) overnight and followed by incubation of the secondary antibody goat anti-rabbit IgG (H+L) 650 (Dylight, 1:400, #84546) and 50 μg/ml DAPI (for nuclear counterstaining), again overnight.

Wash and fixation steps were performed, and xenografts mounted between two coverslips, allowing double side acquisition using Mowiol mounting media (Sigma).

**Statistical analysis**. Statistical analysis was performed using GraphPad Prism 8.0 software. All data sets were challenged by D'Agostino & Pearson and Shapiro–Wilk normality tests. In general, data sets with a Gaussian distribution were analyzed by parametric unpaired $t$ test and data sets that did not pass the normality tests were analyzed by nonparametric unpaired Mann–Whitney test. All were two-sided tests with a confidence interval of 95%. The exception was related to Fig. 4 where data were analyzed by paired tests either by a parametric paired $t$ test (in Fig. 4g) or by paired nonparametric Wilcoxon test (Fig. 4h). Differences were considered significant at $P < 0.05$ and statistical output was represented as follows: non-significant (ns) ≥0.05, *<0.05, **<0.01, ***<0.001, ****<0.0001. The graphs indicate the results as AVG ± standard error of the mean (SEM) or standard deviation (SD). In addition, for small number of samples (<10), we performed an effect size analysis—Cohen's D with a Hedges' g correction ($g$), using Cohen's D 1988 scale ($g$): $g > 0.2$ low; $g > 0.5$ moderate; $g > 0.8$ high.

**Reporting summary**. Further information on research design is available in the Nature Research Reporting Summary linked to this article.

## Data availability
Data from the bulk and single-cell RNA-seq analyses have been deposited in NCBI Gene Expression Omnibus (GEO) with the accession number GSE163751. Source data from Figs. 1–7, and Supplementary Figure 7 are provided as a Source data file. The remaining data are available within the Article, Supplementary Information, or available from the authors upon request. Source data are provided with this paper.

## Code availability
The code to reproduce results from bulk and single-cell RNA-seq is available at Supplementary Software 1.

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

## Acknowledgements

We thank Champalimaud Foundation, LPCC-Terry Fox prize 2018, Congento (LISBOA-01-0145-FEDER-022170, co-financed by FCT/Lisboa2020) for funding. FCT fellowships for V.P. (SFRH/BD/118252/2016), M.M.L. (PD/BD/138203/2018), A.R.G. (CEECIND/02699/2017), and D.S. (PTDC/MED-ONC/28660/2017). We are grateful for M.G. Ferreira's initial support of this project; M.G. Carvalho for co-supervising MD bioinformatic analysis; all members of the Fior Lab for critical discussions; R. Mendes, M. Negrão, and B. Costa for sharing data; the Surgery Digestive Units from Champalimaud Clinical Center (CCC, Dr. N. Figueiredo) and Hospital Professor Fernando Fonseca (HPFF, Dr. V. Nunes and Dr. A. Gomes); The Histopathology Units of CCC (Dr. A. Beltran) and HPFF (Dr. A. Alves); Champalimaud Foundation (CF) Imaging (D. Accardi) and Flow Cytometry Platform (A. Vieira and R. Colaço) for support. We also thank Dr. M. Castillo-Martin, Director of the CF Biobank (CFB), and all the CFB team for human specimens procurement and advice; the CF Fish Platform (C. Certal, J. Monteiro et al.) and Vivarium Rodent Platform (I. Campos) for excellent animal care. We thank M.F. Moraes-Fontes, P. Moura Alves, and D. Henrique for the critical reading of the manuscript; Single Cell Discoveries (M. Muraro) for help with the scRNAseq initial data analysis and the generosity of the zebrafish community for sharing fish (S. Renshaw, F. Djouad, and Z. Wen). Finally, we would like to specially thank Professor Maria de Sousa (1939–2020) for her mentoring, support, enthusiasm, and critical reading of the manuscript.

## Author contributions

R.F. conceptualized and supervised the research; V.P.O., C.R.A., M.M.G., M.M.L., M.F., C.S., and R.F. performed research; M.D. analyzed bulk RNA-seq data; D.S., A.R.G., and R.F. analyzed single-cell RNA-seq data; V.P.O. and R.F. wrote the paper.

## Competing interests

The authors declare no competing interest.
