## [Peer Review File · Nature Communications]

Reviewers' comments:

Reviewer #1 (Remarks to the Author): with expertise in colorectal cancer - mouse models

This is an interesting manuscript reporting observational data on comparison of engraftment for human CRC cell lines in zebrafish embryos. The authors report that tumor engraftment is reduced or "cleared" in two cell lines (regressors) compared with several CRC and TNBC cell lines (progressors). The observation that engraftment of regressor cell lines was enhanced by chemo/radio therapy suggested an immunosuppressive effect.

The manuscript primarily focused on comparison of tumor engraftment in zebrafish embryos of two CRC cell lines derived from the same patient, SW480 (regressor) and SW620 (progressor). The authors suggest that these cell lines illustrate intra-patient heterogeneity and eventually the original immunoediting process from primary to metastasis progression. Comparative transcriptome analysis was performed on a pool of zebrafish xenografts comparing SW480 and SW620 tumors. Hierarchical clustering identified a cluster of 419 differentially expressed genes that upon gene enrichment analysis suggested differences highlighted biologic responses in several areas, including immune response, metabolism and signaling. The prominent difference in hypoxia/angiogenesis were examined previously and the authors focused on differences in immune responses.

The authors present strong evidence to convincingly demonstrate that progressors reduce the clearance of regressors, by using SW480/620 mixed tumors. Macrophage and neutrophil recruitment differ between these groups, and chemical or genetic suppression of these myeloid cells enhances tumor engraftment. Additionally, the data suggest that SW480 cells recruit neutrophils and macrophages more efficiently than SW620, whereas SW620 can polarize macrophages towards a M2-like pro-tumoral phenotype.

The authors conclude their results broaden the application of zebrafish xenografts to function as living biomarkers of the immune response state, that could potentially result in new strategies to enhance immunotherapy.

Overall the manuscript is well written and presented, given the limitations noted below. The manuscript will be of interest to investigators in this field. A principal concern that diminishes the enthusiasm for publication are the data regarding the intrinsic proliferative capacity of the cell lines used for the focused studies. The authors emphasize that tumor engraftment capacities are independent of cell proliferation and apoptosis. However, this statement is based on comparison of the TNBC cell line MDA_MB-468 and CRC cell line Hs578T, both characterized as progressors. Whereas, a prior report (cited as reference 20) characterizing murine xenografts of the two cell lines that constitute the bulk of the data presented reports enhanced proliferative capacity in SW620 (as determined by BrdU incorporation) and enhance apoptosis (TNFalpha induced) in SW480 murine xenografts. This discrepancy should be addressed directly. The comments regarding neutrophil re-education of macrophages are highly speculative at this point and should be de-emphasized. The fact that many of the observations seen in the zebrafish model are not apparent in the mouse xenografts is a major point that should be addressed. This is of concern given that the authors propose using the fast zebrafish model as a biomarker for immune response status in human tumor-derived xenografts. If this zebrafish model does not correlate well with murine models this will diminish the impact of the authors findings.

Several minor concerns

Line 98: This led us to the hypothesis that chemo/radio therapy by eliciting an immunosuppressive effect reduce the zebrafish host anti-tumor response, originally responsible for the regressor behavior.

This is an awkward statement. Perhaps the authors are suggesting that chemo/radio therapy selects for a tumor cell population that results in reduction of the zebrafish host anti-tumor response. Alternatively, are they suggesting that chemo/radio therapy directly modulates the host

anti-tumor response?

Line 118: "Genes with a False Discovery Rate (FDR) < 0.06 and fold change > 1 were considered significant." The authors have used a fold change of >1 as a cut off, so basically any change that is highlighted as highly significant. If accepting this level of change, then an effect size test is important to measure the magnitude of the occurring fold change (such as Cohen's d test). This will help readers gauge how much of a change actually occurs in differentially expressed genes, not just whether they hit an arbitrary significance.

Heatmap data in Figure 1d are vague and uninformative. For example:

What do the values represent? Fold change? Are these data normalized reads? Is it n=4 vs n=3 for the analysis? Would a slightly larger fold change cut off (such as 1.2 FC, 0.06 FDR) for this data be more informative? Depending on what your values are in the heatmap, consider Pearson's coefficient instead of Euclidean clustering for the genes as this can be highly informative.

Supplementary figure 3. The authors conclusion that recruitment is independent of tumor size (no correlation between tumor size and innate immune cell recruitment) is somewhat overstated. While the SW480 vs neutrophils show no correlation, the SW420 tumor cell number correlation with macrophages number maybe better characterized as modest correlation ($R^2=0.52$ for in vivo data), and the other may be better characterized as modest to weak correlation.

Reviewer #2 (Remarks to the Author): expertise in Zebrafish

Key Results:

In this manuscript the authors examine the engraftment of colorectal cells in a zebrafish larval xenograft model. They demonstrate that two different colorectal cell lines (SW480 and SW620 - both isolated from the same patient) display differing engraftment rates. The SW480 cells have a low engraftment efficiency and compared to SW620 cells. The authors hypothesise that the differing engraftment profiles is caused by the SW620 cells "evading" the innate immune response of the host. In support of this, they show that SW480 cells recruit higher levels of innate immune cells, especially M1-type macrophages and that loss of macrophages, using either haemopoietic mutants or by clodronate-mediated ablation, is able to enhance SW480 cell engraftment. In addition, they show that SW620 cells can protect SW420 cells in heterologous grafts. They then perform some mouse xenograft work which shows that SW480 cells also recruit higher levels of M1 macrophages and grow better following macrophage ablation.

Originality, significance, validity and conclusions:

Overall this is an interesting study which demonstrates a clear role for macrophages in xenograft stability. The concept that innate immune cells can contribute towards immunoediting in tumours is relatively novel and the authors provide some elegant data that suggests this is happening in zebrafish xenografts. The fact that immune silenced tumour cells can somehow protect susceptible tumour cells from the innate immune response is also novel but the mechanism by which this occurs is not explored here. However, it is difficult to see how general conclusions can be drawn on native tumour/immune interactions given that nearly all of the data is generated in zebrafish xenografts.

Zebrafish xenografts are likely to induce a different immune response than those induced by a tumour in a human patient as there are species differences between the tumour and the tumour micro-environment (human cells in a zebrafish host), differences in temperature (34 vs 37 degrees) and because the site of tumour engraftment (perivitelline space) is clearly different from

the human colon. Because of these disparities, it is difficult to determine how relevant this study is to human cancer and to me this therefore reduces the impact of this study in its current form. In addition while the immunoediting data presented in Figure 7 is exciting, further analysis should be conducted on these escapers to show that "editing has indeed taken place.

If the authors were to provide more convincing data that the mechanisms of innate immune silencing are conserved and are not just specific to fish xenografts and provide more evidence of tumour transcriptional changes following "immunoediting" then I believe this study would appeal to a wider audience and have more impact worthy of publishing in Nature Communications. Such evidence might include:

- further and more detailed analysis of mouse xenografts.
- analysis of non-xenografted tumours. For example, using a mouse genetic model of colon cancer.
- further RNA-Seq analysis of edited and non-edited tumour cells.
- transplantation of "immunoedited" cells into syngeneic hosts

Statistics:

Overall the statistics used are appropriate. However, the standard deviation should be used to display the variance in graphs instead of SEM.

Referencing:

Appropriate.

Clarity and context

Generally, well written.

Suggested Improvements

1) Ideally an assay that does not rely on xenotransplantation should be used. Consider the use of a genetic or mutagen-induced murine cancer model to generate tumours and then investigate tumour growth in the absence of macrophages using clodronate. In addition, to demonstrate immunoediting the "immunoedited" tumours (i.e. those that escape immune clearance) should be able to survive when implanted into a syngeneic host.

2) The mouse studies are a good start and generally show similar results but I was concerned that the engraftment rates were identical between 480 and 620 cells in mice xenografts, this suggests to me that there might be different mechanisms at play between the immune silencing in mice and zebrafish xenografts.

Further experiments to show conservation of immune silencing between fish and mouse hosts are required and could include:

- macrophage depletion in 620 grafts in mice. These should show little change in engraftment/tumour volume as these cells are already "immune silent"
- macrophage depletion in 480/620 "immune silent" grafts in mice.
- Presumably macrophage-depleted mouse hosts should allow other xenografts to progress quicker? For example, xenografts of the other regressors identified in Figure 1.
- While engraftment rates were unchanged between 480 and 620 grafts in mice it would be useful to include data on tumour size in mice xenografts – did 620 grafts grow better than 480 grafts?

3) The RNA-Seq analysis compared SW480 to SW620 cells isolated from 2 dpi xenografts, however to truly show "immunoediting" the authors should demonstrate differences in transcriptional profiles before and after engraftment. For example were the "immune-related" pathways already enriched in SW480 cells prior to engraftment or were they upregulated following xenotransplantation?

- Alternatively could the authors compare transcriptional profiles of SW480 cells in a 50:50 SW480/620 mix (by FACs) compared to homogenous xenografts?

4) It is not clear what the authors define as "engraftment" vs non-engraftment. Presumably even a few labeled cells at 4 dpi is defined as grafted or was there a minimum cutoff?

5) The studies used to determine that differences in engraftment potential were not related to cell growth or apoptosis were only conducted in HCT116 and MDA-MB-468 cells and should be extended to include SW480 and SW620 cells which were the main focus of this study. This is especially important as previous studies have highlighted the susceptibility of SW480 to apoptosis and the increased cell proliferation in SW620 in serum starved culture. (See Hewitt et al., Reference #20 in manuscript).

6) There is no data demonstrating the efficacy and specificity of clodronate knockdown on overall macrophage or neutrophil numbers in mice or fish. This should be included.

7) The authors claim that both macrophages and neutrophils are required for immune editing and have based this on analysis of two mutants *runx1* (loss of neutrophils) and *csf1ra* (loss of macrophages). Data should be included to confirm the specificity of these mutants in the depletion of either neutrophils or macrophages. This is especially important as the *runx* mutant used may actually increase macrophage number which could confound the analysis. (see <https://doi.org/10.1182/blood-2011-12-398362>). Also, the papers cited on the effects of these two mutants generally analyse innate immune cell numbers earlier in development than is the case here (up to 6 dpf) and need validating at the time points used in this study.

8) Related to point 7, to exclude a role for *runx1* in macrophage function, the authors should specifically ablate neutrophils to confirm the role of these innate immune cells – I suggest a nitroreductase or similar approach is used.

9) Are mixed 480/620 are grafts able to show the same effect on biasing macrophage polarisation that they do on immune cell recruitment?

10) The immunoediting of SW480 cells is interesting but this experiment would be strengthened by the incorporation of a few controls:

- Use of SW620 cells which presumably would show no improvement despite selection from 4 dpi larvae and passaging.

- Analysis of cell proliferation and apoptosis in SW480 escapers to determine if phenotype is not simply due to the selection of more proliferative cells from the original SW480 culture.

11) It would be interesting to determine how macrophages are driving the loss of SW480 cells in grafts. Is this through induction of apoptosis followed by phagocytosis of dying cells? Is the rate of apoptosis reduced in SW480 cells following macrophage ablation or in mixed 480/620 grafts?

12) The zebrafish mutants should be italicised.

Reviewer #3 (Remarks to the Author): expertise in Zebrafish - tumor models

In this manuscript Póvoa et al. present some excellent work demonstrating that the innate immune effector cells of the zebrafish embryo respond differently to different xenografted human tumour cells, including two cell lines derived from the same patient, representing primary and metastasized tumour cells. They use the power of the zebrafish xenograft system to dissect the roles of macrophages vs. neutrophils in tumour clearance, and show convincingly that engraftment efficiency correlates strongly with evading the innate immune effector cells of the host, and furthermore, that the macrophages responding to efficiently engrafting cells are expressing significantly different suites of genes. This not only provides valuable insight into the roles of innate immune effector cells in the tumour microenvironment, but also helps develop the zebrafish xenograft platform as a powerful system in which the mechanisms of macrophage polarization in the TME can be more directly observed.

The manuscript is largely well organized, complete, and well-reasoned; there are a lot of minor English language issues and formatting/typographic problems, but it is not at all difficult to understand. The data are of good quality, and largely support the conclusions drawn. I have only 1 substantive issue with the manuscript in its current form, as well as a few minor and trivial quibbles, all of which can be easily addressed.

Major:

1) The title is misleading. The experimental results do not reveal the status of the human tumour microenvironment; they allow the researcher to experimentally manipulate the zebrafish TME and directly image and otherwise readily assay the effects on the tumour biology, which is part of the power of the zebrafish xenograft model. In fact, the authors effectively state as much at the end of their introduction (line 69) "Here we hypothesize that the zebrafish innate immune system can be modulated by a tumor, itself capable of generating an immunosuppressive TME or subjected to elimination." The authors need to provide a more accurate and informative title.

Minor

1) Line 236; Using the runx/panther mutants is an elegant approach to looking at the effects of neutrophil/macrophage depletion, but these are definitely indirect, rather than direct tests of these hypotheses; both mutations may be having many other effects that alter the biology of the TME, most notably changes in MMP expression and/or activation.

The issue of differences between host and tumour MMP suite has been recently reviewed (see Wyatt, et al. 2017 "Zebrafish Xenograft: An Evolutionary Experiment in Tumour Biology." *Genes*, 8(9), 220. <http://doi.org/10.3390/genes8090220> - it might be worth adding this reference). This is also relevant WRT the issue of using the zebrafish, vs. murine xenograft models; it was very gratifying to see this issue addressed directly and conclusively in this study.

2) Imaging - What lens (what NA)? The 5 μm step size seems awfully coarse; was PH diameter increased such that the FWHM was $> 5 \mu\text{m}$? Was this necessary due to extremely weak signals? Given that these were fixed specimens, I don't understand why a greater sampling frequency (smaller step size) was not used? This probably dramatically under-sampled the axial resolution of the system. I'm not certain, but the fact that the data therefore represents (possibly non-random) subsamples of the experimental populations may invalidate some of the assumptions of the statistical analyses.

3) The authors should provide a spreadsheet with all 419 DEGs as a supplemental file; I am very

curious about the specific genes you identified.

Trivial

1) Minor English language issues throughout; needs proof reading.

2) figure 2f and comparable figures: y-axis "tumour size" is presumably tumour volume? Add units (presumably μm^3).

Finally, just as a comment, the experiment comparing the engraftment of 'SW480zEscapers' is very similar to one that I've had in mind for a few years (but have not been able to find the resources to do); it would be very interesting to compare the transcriptome of escapers to the parental SW480 cells; especially after two or more rounds of xenografting and recovery.

Similarly, the authors allude to the 'amazing plasticity and diversity of macrophage populations' in particular; it would be very interesting to use the zebrafish xenograft model to compare the transcriptomes of the tumour-permissive macrophages associated with progressing tumours to those associated with tumour clearance. ^[1]_{SEP}

I look forward to seeing this, and the future work from this group published.

REVIEWER #1

(Remarks to the Author): with expertise in colorectal cancer - mouse models

We thank Reviewer#1 for carefully reading our manuscript and raising important concerns that we will try to address point by point.

This is an interesting manuscript reporting observational data on comparison of engraftment for human CRC cell lines in zebrafish embryos. The authors report that tumor engraftment is reduced or “cleared” in two cell lines (regressors) compared with several CRC and TNBC cell lines (progressors). The observation that engraftment of regressor cell lines was enhanced by chemo/radio therapy suggested an immunosuppressive effect.

The manuscript primarily focused on comparison of tumor engraftment in zebrafish embryos of two CRC cell lines derived from the same patient, SW480 (regressor) and SW620 (progressor). The authors suggest that these cell lines illustrate intra-patient heterogeneity and eventually the original immunoeediting process from primary to metastasis progression. Comparative transcriptome analysis was performed on a pool of zebrafish xenografts comparing SW480 and SW620 tumors. Hierarchical clustering identified a cluster of 419 differentially expressed genes that upon gene enrichment analysis suggested differences highlighted biologic responses in several areas, including immune response, metabolism and signaling. The prominent difference in hypoxia/angiogenesis were examined previously and the authors focused on differences in immune responses.

The authors present strong evidence to convincingly demonstrate that progressors reduce the clearance of regressors, by using SW480/620 mixed tumors. Macrophage and neutrophil recruitment differ between these groups, and chemical or genetic suppression of these myeloid cells enhances tumor engraftment. Additionally, the data suggest that SW480 cells recruit neutrophils and macrophages more efficiently than SW620, whereas SW620 can polarize macrophages towards a M2-like pro-tumoral phenotype.

The authors conclude their results broaden the application of zebrafish xenografts to function as living biomarkers of the immune response state, that could potentially result in new strategies to enhance immunotherapy.

Overall the manuscript is well written and presented, given the limitations noted below. The manuscript will be of interest to investigators in this field.

1. A principal concern that diminishes the enthusiasm for publication are the data regarding the **intrinsic proliferative capacity of the cell lines used for the focused studies**. The authors emphasize that tumor engraftment capacities are independent of cell proliferation and apoptosis. However, this statement is based on comparison of the TNBC cell line MDA_MB-468 and CRC cell line Hs578T, both characterized as progressors. Whereas, a prior report (cited as reference 20) characterizing murine xenografts of the two cell lines that constitute the bulk of the data presented reports enhanced proliferative capacity in SW620 (as determined by BrdU incorporation) and enhance apoptosis (TNFalpha induced) in SW480 murine xenografts. This discrepancy should be addressed directly.

We thank reviewer#1 for bringing this to our attention and we now include in Supplementary Figure1 the analysis and comparison of proliferation and apoptosis in SW480 and SW620 zebrafish xenografts at 2 time points: 1day and 4 days post injection. Indeed, at 1dpi we could detect a higher absolute number and percentage of mitosis in SW620 (**Supplementary Figure 1c-c'**). However, at 4dpi the tumors become very big and although in absolute numbers this difference maintains, if we take in account the

total number of cells, the percentage of proliferating cells is reduced in SW620 in relation to SW480.

As for apoptosis, at 1dpi we could not detect major differences between the 2 types of xenografts but at 4dpi SW620 present a higher percentage of apoptotic cells (**Supplementary Figure 1d-d'**). In summary proliferation and apoptosis is higher in SW620 than SW480. Thus, if from one side proliferation of SW620 may indeed correlate with engraftment, apoptosis does not. Therefore, we believe that proliferation and apoptosis are not the main drivers of clearance / engraftment.

2- The comments regarding neutrophil re-education of macrophages are highly speculative at this point and should be de-emphasized.

We thank reviewer#1 for your comment and we now tone down our claims.

3-The fact that many of the observations seen in the zebrafish model are not apparent in the mouse xenografts is a major point that should be addressed. This is of concern given that the authors propose using the fast zebrafish model as a biomarker for immune response status in human tumor-derived xenografts. If this zebrafish model does not correlate well with murine models this will diminish the impact of the authors findings.

We thank reviewer#1 for your comment and we now try to enrich our discussion: please see lines 286-291 and 300-306.

Indeed, SW480/SW620 mouse Rag1^{-/-} C57BL6/N xenografts did not show major differences in engraftment. However, we re-analyzed the data and in fact there was one SW480 tumor (1 out of 5 injected), that when analyzed for flow cytometry had no presence of human cells (0,016%GFP cells) (**Figure R1**- dot in blue represents this SW480 tumor/xenograft). In contrast in SW620 all 5 injected mice had the presence of human cells (>80% of human cells). At the time we dismissed these results since we though 1 out 5 could be just a technicality but may indicate that engraftment was not as efficient as SW620.

Importantly, in the previous mouse paper by *Hewitt et al, 2000*, the original characterization of the pair of cell lines, authors clearly show a difference in engraftment (Fig. 3C, ~20 mice, at 1x10⁶ injected cells) in the same line as we observed in zebrafish. However, instead of using mouse Rag1^{-/-} C57BL6/N, they used BalbC nude mice, which to our understanding can generate a lot of discrepancies in results, not only due to the underlying mutations but also the background strains.

We did not repeat these experiments due to COVID19 restrictions and ethical reasons. According to our Institutional Animal Welfare Organ (ORBEA, in the Portuguese acronym), "repeating these experiments would not be in-line with the best practices for animal experimentation, specifically as it would require the use of a considerable number of animals with a very marginal benefit, in essence, the confirmation of an already published result." The execution of this experiment and use of additional animals for this purpose, is also made more difficult in the current context of the Animal House, operating under severe COVID19 contingencies with animal colony size reductions, staff shortages and users restrictions.

Nevertheless, analysis of the ratios of the different macrophage populations F4/80⁺CD80⁺ macrophages, showed that SW480 tumors were enriched in “M1-like” anti-tumoral macrophage population in relation to SW620 xenografts (**Fig. 6a-c, d**, $P=0.0159$), similar to our zebrafish experiments. Also, like in zebrafish, SW620 cells became dominant in the MIX mouse xenografts (**Fig. 6e**, $*P=0.0286$).

As a general comment on the differences of the models, the analysis of the murine model was performed ~21 days post tumor injection and our zebrafish TME analysis at 1 and 4 days post injection, i.e. a discrepancy of 20 days. We believe that the zebrafish model allows for an immediate quick snapshot of the “tumor state” but the murine model allows to study how this tumor-TME interactions evolve along time, therefore this “timing” issue can really account for a lot of differences, and does not necessarily undermine one model or the other.

Several minor concerns

4-Line 98: This led us to the hypothesis that chemo/radio therapy by eliciting an immunosuppressive effect reduce the zebrafish host anti-tumor response, originally responsible for the regressor behavior.

This is an awkward statement. Perhaps the authors are suggesting that chemo/radio therapy selects for a tumor cell population that results in reduction of the zebrafish host anti-tumor response. Alternatively, are they suggesting that chemo/radio therapy directly modulates the host anti-tumor response?

We apologize if we were not clear, but we are indeed suggesting that chemo/radio therapy directly modulates the host anti-tumor response.

We changed the sentence to:

“This led us to the hypothesis that chemo/radio therapy elicits an immunosuppressive effect, reducing the zebrafish host anti-tumor response, originally responsible for clearance.”

It has been shown that conventional cytotoxic chemotherapy may suppress the hematopoietic system, leading many times to neutropenia, myelopenia or even lymphopenia (Rasmussen & Arvin, 1982, doi: [10.1289/ehp.824321](https://doi.org/10.1289/ehp.824321), PMID: [7037385](https://pubmed.ncbi.nlm.nih.gov/7037385/), (Vitale et al., 2011, PMID: [26678337](https://pubmed.ncbi.nlm.nih.gov/26678337/), doi: [10.1016/j.ccell.2015.10.012](https://doi.org/10.1016/j.ccell.2015.10.012)). In fact, we analyzed the TME of SW480 upon FOLFIRI treatment and observed a reduction in neutrophil infiltrate (see **Figure R2**).

5-Line 118: “Genes with a False Discovery Rate (FDR) < 0.06 and fold change > 1 were considered significant.” The authors have used a fold change of >1 as a cut off, so basically any change that is highlighted as highly significant. If accepting this level of change, then an effect size test is important to measure the magnitude of the occurring fold change (such as Cohen’s d test). This will help readers gauge how much of a change actually occurs in differentially expressed genes, not just whether they hit an arbitrary significance.

We thank reviewer #1 for raising this point. There is a typo in the first sentence as we have considered “Genes with a False Discovery Rate (FDR) < 0.07 and absolute log₂ fold-change > 1”. Indeed, we have decided to apply a more stringent significance threshold. More than an adjusted p-value cutoff, we required absolute log₂ fold-changes to be above a minimum value, greater than 1, which is equivalent to a 2-fold difference between the SW480 and SW620 xenografts on the original scale.

6-Heatmap data in Figure 1d are vague and uninformative. For example: What do the values represent? Fold change? Are these data normalized reads? Is it n=4 vs n=3 for the analysis? Would a slightly larger fold change cut off (such as 1.2 FC, 0.06 FDR) for this data be more informative? Depending on what your values are in the heatmap, consider Pearson’s coefficient instead of Euclidean clustering for the genes as this can be highly informative.

We thank reviewer #1 for raising this point and apologize for not being clear.

The Heatmap in Figure 1d presents a two-dimensional dendrogram (based on Pearson’s correlation coefficient distance) of log₂ counts-per-million (logCPM) normalized expression values of differentially expressed genes (N = 459) in SW480 (low engraftment) versus SW620 (high engraftment) comparison, where colors represent expression values scaled by row (Z-scores). As referred in previous point, the genes were selected using a stringent cut-off of FDR < 0.06 and absolute logFC > 1. We completed the figure legend with all this information.

7-Supplementary figure 3. The authors conclusion that recruitment is independent of tumor size (no correlation between tumor size and innate immune cell recruitment) is somewhat overstated. While the SW480 vs neutrophils show no correlation, the SW420 tumor cell number correlation with macrophages number maybe better characterized as modest correlation (R²=0.52 for in vivo data), and the other may be better characterized as modest to weak correlation.

We thank reviewer #1 for raising this point. In relation to SW480 number of cells / macrophage correlation indeed has a R² of 0,52, however if we take out the highest point (an outlier determined by GraphPad calculator), the R² becomes 0.10, thus we have corrected the graph (see **Supplementary Figure 3**) and the text to modest to week correlation.

REVIEWER #2

(Remarks to the Author): expertise in Zebrafish

We thank Reviewer#2 for carefully reading our manuscript and raising important concerns that we will try to address point by point.

Key Results:

In this manuscript the authors examine the engraftment of colorectal cells in a zebrafish larval xenograft model. They demonstrate that two different colorectal cell lines (SW480 and SW620 - both isolated from the same patient) display differing engraftment rates. The SW480 cells have a low engraftment efficiency and compared to SW620 cells. The authors hypothesise that the differing engraftment profiles is caused by the SW620 cells “evading” the innate immune response of the host. In support of this, they show that SW480 cells recruit higher levels of innate immune cells, especially M1-type macrophages and that loss of macrophages, using either haemopoietic mutants or by clodronate-mediated ablation, is able to enhance SW480 cell engraftment. In addition, they show that SW620 cells can protect SW420 cells in heterologous grafts. They then perform some mouse xenograft work which shows that SW480 cells also recruit higher levels of M1 macrophages and grow better following macrophage ablation.

Originality, significance, validity, and conclusions:

Overall this is an interesting study which demonstrates a clear role for macrophages in xenograft stability. The concept that innate immune cells can contribute towards immunoediting in tumours is relatively novel and the authors provide some elegant data that suggests this is happening in zebrafish xenografts. The fact that immune silenced tumour cells can somehow protect susceptible tumour cells from the innate immune response is also novel but the mechanism by which this occurs is not explored here. However, it is difficult to see how general conclusions can be drawn on native tumour/immune interactions given that nearly all of the data is generated in zebrafish xenografts.

Zebrafish xenografts are likely to induce a different immune response than those induced by a tumour in a human patient as there are species differences between the tumour and the tumour micro-environment (human cells in a zebrafish host), differences in temperature (34 vs 37 degrees) and because the site of tumour engraftment (perivitelline space) is clearly different from the human colon.

Because of these disparities, it is difficult to determine how relevant this study is to human cancer and to me this therefore reduces the impact of this study in its current form. In addition while the immunoediting data presented in Figure 7 is exciting, further analysis should be conducted on these escapers to show that “editing has indeed taken place.

If the authors were to provide more convincing data that the mechanisms of innate immune silencing are conserved and are not just specific to fish xenografts and provide more evidence of tumour transcriptional changes following “immunoediting” then I believe this study would appeal to a wider audience and have more impact worthy of publishing in Nature Communications. Such evidence might include:

- further and more detailed analysis of mouse xenografts
- analysis of non-xenografted tumours. For example, using a mouse genetic model of colon cancer.
- further RNA-Seq analysis of edited and non-edited tumour cells
- transplantation of “immunoedited” cells into syngeneic hosts

Statistics:

Overall the statistics used are appropriate. However, the standard deviation should be used to display the variance in graphs instead of SEM.

We thank the reviewer#2 for the suggestion and altered all the columns graphs to STDEV. The other graphs (dots-each dot a xenograft) the dispersion is clear and therefore we do not see the advantage of displaying it, since the graph perception will be distorted, so we maintained SEM.

Referencing: Appropriate.

Clarity and context

Generally, well written.

Suggested Improvements

1) Ideally an assay that does not rely on xenotransplantation should be used. Consider the use of a genetic or mutagen-induced murine cancer model to generate tumours and then investigate tumour growth in the absence of macrophages using clodronate. In addition, to demonstrate immunoediting the “immunoedited” tumours (i.e. those that escape immune clearance) should be able to survive when implanted into a syngeneic host.

We thank the reviewer#2 for the suggestion, however we did not have conditions to perform these experiments in mice. We do not have a license to perform these experiments, thus we would have to ask a new licence to our Institutional Animal Welfare Organ (ORBEA, in the Portuguese acronym) and then to the Portuguese National Authority for Animal Health (DGAV – Portuguese acronym), this would not be compatible with the review timings. Also, in the current context of COVID19, the Animal House is operating under contingencies with animal colony size reductions and staff shortages and users restrictions.

2) The mouse studies are a good start and generally show similar results but I was concerned that the engraftment rates were identical between 480 and 620 cells in mice xenografts, this suggests to me that there might be different mechanisms at play between the immune silencing in mice and zebrafish xenografts.

Further experiments to show conservation of immune silencing between fish and mouse hosts are required and could include:

- macrophage depletion in 620 grafts in mice. These should show little change in engraftment/tumour volume as these cells are already “immune silent”
- macrophage depletion in 480/620 “immune silent” grafts in mice.
- Presumably macrophage-depleted mouse hosts should allow other xenografts to progress quicker? For example, xenografts of the other regressors identified in Figure
- While engraftment rates were unchanged between 480 and 620 grafts in mice it would be useful to include data on tumour size in mice xenografts – did 620 grafts grow better than 480 grafts?

We thank Reviewer#2 for the comments and suggestions. Please see answer to comment #3 also raised by Reviewer#1.

3) The RNA-Seq analysis compared SW480 to SW620 cells isolated from 2 dpi xenografts, however to truly show “immunoediting” the authors should demonstrate differences in transcriptional profiles before and after engraftment. For example were the “immune-related” pathways already enriched in SW480 cells prior to engraftment or were they upregulated following xenotransplantation? Alternatively could the authors compare transcriptional profiles of SW480 cells in a 50:50 SW480/620 mix (by FACs) compared to homogenous xenografts?

We thank reviewer#2 for the suggestion and indeed we performed a similar experiment to find the escapers clones. We performed single cell RNAseq of SW480 xenografts at 2 time points: 1dpi and 4dpi. From 1dpi to 4dpi, some tumors clones disappear (get cleared) and others remain (escapers – the engrafted ones). In this way we were able to

identify the clones that get cleared (are no longer represented at 4dpi) and also the ones that remain and get expanded. Please see new Figures 8 and 9 and Supplementary Figures 7 and 8 and corresponding text.

4) It is not clear what the authors define as “engraftment” vs non-engraftment. Presumably even a few labeled cells at 4 dpi is defined as grafted or was there a minimum cutoff?

We apologize if we were not clear. Our threshold is N=30 cells – below 30 we consider No tumor, and above this we consider that we have engraftment. We have now included this information in the text.

5) The studies used to determine that differences in engraftment potential were not related to cell growth or apoptosis were only conducted in HCT116 and MDA-MB-468 cells and should be extended to include SW480 and SW620 cells which were the main focus of this study. This is especially important as previous studies have highlighted the susceptibility of SW480 to apoptosis and the increased cell proliferation in SW620 in serum starved culture. (See Hewitt et al., Reference #20 in manuscript).

We thank reviewer#2 for bringing this point. Please see answer to comment number 1 also raised by Reviewer#1.

6) There is no data demonstrating the efficacy and specificity of clodronate knockdown on overall macrophage or neutrophil numbers in mice or fish. This should be included.

We thank reviewer#2 for bringing this to our attention and we now include a figure where we quantify the number of macrophage and neutrophils in zebrafish transgenics upon L-clodronate treatment. Please see new **Supplementary Figure 6**.

We did not repeat experiments in mice due to COVID19 restrictions as mentioned before. L-clodronate is used as a gold standard in mice experiments to reduce macrophages (Weisser, van Rooijen, & Sly, 2012, PMID: 22871793, DOI: 10.3791/4105). Also, there are several publications describing the effect of L-clodronate in the innate immune populations (Van Rooijen & Sanders, 1994), PMID: 8083541, DOI: 10.1016/0022-1759(94)90012-4).

7) The authors claim that both macrophages and neutrophils are required for immune editing and have based this on analysis of two mutants *runx1* (loss of neutrophils) and *csf1ra* (loss of macrophages). Data should be included to confirm the specificity of these mutants in the depletion of either neutrophils or macrophages. This is especially important as the *runx* mutant used may actually increase macrophage number which could confound the analysis. (see <https://doi.org/10.1182/blood-2011-12-398362>). Also, the papers cited on the effects of these two mutants generally analyse innate immune cell numbers earlier in development than is the case here (up to 6 dpf) and need validating at the time points used in this study.

We thank reviewer#2 for bringing this to our attention and we have now quantified both populations in *runx1^{w84x}* and *csf1ra^{i4blue}*, at 3dpf and 6dpf (**Figure R3**).

Given the fact that we do not have an antibody that consistently labels the macrophage population we had to backcrossed our mutants with a macrophage reporter (*mpeg:dsRED* or *fms:GFP*) to quantify macrophages. To quantify neutrophils, in the same fish, we can perform immunofluorescence for *mpx* with the GENETEX anti-*mpx* antibody and in this way quantified both populations.

However, during this process of review/COVID lockdown, unfortunately our *runx1^{w84x/-}* homozygous colony died so we had to use the heterozygous *runx1^{w84x +/-}* fish. We in-crossed these heterozygous *runx1^{w84x +/-}* fish and quantified the different populations and then sort them according to the putative mendelian distribution.

Since it is described that *runx1^{w84x/-}* have low numbers of neutrophils, we ranked the fish according to the neutrophil distribution in quarters – lower quarter would correspond to the putative *runx1^{w84x}* homozygous, second and 3rd quarter to heterozygous and 4th quarter with higher number of neutrophils would correspond to the Wt siblings. Then, we quantified the macrophages in each of these quarters (i.e. defined by the neutrophil ranking). In other words, in graphs a/b and c/d pairs, each column in graph a have the same fish as graph b. The same for c-d chart.

We had some fixed *runx1^{w84x/-}* mutant fish and *wt* from earlier experiments at 6dpf, that we quantified (**Figure. R3**) but not from the 3dpf time point. In *runx1^{w84x}* we were able to observe lower numbers of neutrophils at 3dpf and 6dpf – however, we could not observe at these time points the described compensation of macrophages.

Figure R3 |Quantification of neutrophils and macrophages in *runx1^{w84x}* and *csfr1a^{4blue}*.
a, c. In-cross of heterozygous *runx1^{w84x}* fish and quantification of neutrophils (whole body) sorted according the putative mendelian distribution. **b, d.** Macrophages quantification in each of these quarters (i.e. defined by the previous neutrophil ranking). In other words, in graphs a/b and c/d pairs, each column in graph a have the same fish as graph b. The same for c-d chart. **e.** Quantification of neutrophils at 6dpf in *runx1^{w84x/-}* and WT zebrafish.
f, h. In-cross of heterozygous *csfr1a^{4blue}* fish and quantification of macrophages (whole body), sorted according the putative mendelian distribution. Neutrophils quantification in each of these quarters (i.e. defined by the previous macrophage ranking). In other words, in graphs f/g and h/i pairs, each column in graph f have the same fish as graph g. The same for h-i chart. Each dot represents a zebrafish embryo. Embryos are from one clutch. N is depicted in the charts. Error bars indicate mean ± SEM. ****P<0.0001

In Panther *csfr1a^{4blue}* we had to apply the same strategy, i.e. cross heterozygous and since panther mutants have lower numbers of macrophages we ranked the fish according to the macrophage distribution in quarters – lower quarter would correspond to the putative Panther homozygous, second and 3rd quarter to heterozygous and 4th quarter with higher number of macrophages would correspond to the Wt siblings. Indeed, we could observe a population with lower number of macrophages but again we could not observe the any sort of compensation in the number of neutrophils.

In addition, we found in the literature a thorough quantification of macrophages and neutrophils in the Panther *csfr1a^{4blue}* at 3dpf and 6dpf – Figure S2 (Pagán, et al. DOI: 10.1016/j.chom.2015.06.008). Authors show a consistent decrease in macrophage numbers but no alteration in neutrophils.

8) Related to point 7, to exclude a role for runx1 in macrophage function, the authors should specifically ablate neutrophils to confirm the role of these innate immune cells – I suggest a nitroreductase or similar approach is used.

We thank reviewer#2 for this suggestion, however we were not able to import this transgenic line and perform these experiments.

9) Are mixed 480/620 are grafts able to show the same effect on biasing macrophage polarisation that they do on immune cell recruitment?

We thank reviewer#2 for this suggestion and we now show these quantifications in the new Figure 4. As you can observe, interestingly the MIX xenografts show again a very similar dynamic to SW620 xenografts (more M2 than M1-like macrophages).

10) The immunoediting of SW480 cells is interesting but this experiment would be strengthened by the incorporation of a few controls:

- Use of SW620 cells which presumably would show no improvement despite selection from 4 dpi larvae and passaging.

We did not perform this experiment since engraftment is already very close to 100%.

- Analysis of cell proliferation and apoptosis in SW480 escapers to determine if phenotype is not simply due to the selection of more proliferative cells from the original SW480 culture.

We thank reviewer#2 for this suggestion and we now show these quantifications in the new Figure 7. In fact, we did not detect a higher proliferation rate in escaper clones (Figure 7f), but we did detect higher apoptosis (Figure 7g).

Interestingly in the new scRNAseq experiments we can observe that the clone that expands mostly (cluster#3) is not enriched in proliferation signatures (mitotic cell cycle/prometaphase (Figure 8).

11) It would be interesting to determine how macrophages are driving the loss of SW480 cells in grafts. Is this through induction of apoptosis followed by phagocytosis of dying cells? Is the rate of apoptosis reduced in SW480 cells following macrophage ablation or in mixed 480/620 grafts?

We agree with reviewer#2 and we tried to investigate this but was not very conclusive.

1-Analysis of SW480 apoptosis upon L-clodronate (**Figure R4a**) shows an increase in apoptosis, and in panther mutants (**FigureR4b**) we observe a tendency to have higher apoptosis. This increase in apoptosis is probably due to lack of clearance and to our view does not necessarily exclude your hypothesis of induction of apoptosis, but at this point we are not being able to disentangle these 2 events.

Figure R4 | Apoptosis of SW480 tumors increase in the absence of macrophages. a. Quantification of apoptotic tumor cells (%) upon L-clodronate treatment at 4dpi in SW480 xenografts (Mann Whitney test ns=0.35, **P=0.001). N is depicted in the chart and correspond to 1 independent experiment. b. Quantification of apoptotic tumor cells (%) in *csf1ra*^{fl/fl} (*panther*) mutants at 4dpi in SW480 xenografts (Mann Whitney test ns=0.50). N is depicted in the chart and correspond to 3 independent experiments. c. Quantification of apoptosis in SW480 and SW620 tumors when alone or mixed at 1dpi and 4dpi (Mann Whitney test ns=0.66, ns=0.31, *P=0.021, ns=0.13, ***P=0.0006). N at 1dpi: SW480 N=13, SW620 N=19, SW480/SW620@MIX N=5. N at 4dpi: SW480 N=32, SW620 N=39, SW480/SW620@MIX N=7 and corresponds to 1 independent experiment. Each dot represents a xenograft. Error bars indicate mean ± SEM.

2-Analysis of SW480 apoptosis in MIX xenografts show a tendency to reduce apoptosis in the presence of SW620, possibly reflecting a “protective” role of SW620 – please see **Figure R4c**. In fact, if we take in account that SW620 in MIX seems to reduce neutrophil and macrophage infiltrate and SW480 decrease their apoptosis, the hypothesis is indeed valid.

12) The zebrafish mutants should be italicized

We thank reviewer#2 for pointing this out and have now corrected everything.

REVIEWER #3

(Remarks to the Author): expertise in Zebrafish - tumor models

We thank Reviewer#3 for carefully reading our manuscript and raising important concerns that we will try to address point by point.

In this manuscript Póvoa et al. present some excellent work demonstrating that the innate immune effector cells of the zebrafish embryo respond differently to different xenografted human tumour cells, including two cell lines derived from the same patient, representing primary and metastasized tumour cells. They use the power of the zebrafish xenograft system to dissect the roles of macrophages vs. neutrophils in tumour clearance, and show convincingly that engraftment efficiency correlates strongly with evading the innate immune effector cells of the host, and furthermore, that the macrophages responding to efficiently engrafting cells are expressing significantly different suites of genes. This not only provides valuable insight into the roles of innate immune effector cells in the tumour microenvironment, but also helps develop the zebrafish xenograft platform as a powerful system in which the mechanisms of macrophage polarization in the TME can be more directly observed. The manuscript is largely well organized, complete, and well-reasoned; there are a lot of minor English language issues and formatting/typographic problems, but it is not at all difficult to understand. The data are of good quality, and largely support the conclusions drawn. I have only 1 substantive issues with the manuscript in its current form, as well as a few minor and trivial quibbles, all of which can be easily addressed.

Major:

1) The title is misleading. The experimental results do not reveal the status of the human tumour microenvironment; they allow the researcher to experimentally manipulate the zebrafish TME and directly image and otherwise readily assay the effects on the tumour biology, which is part of the power of the zebrafish xenograft model. In fact, the authors effectively state as much at the end of their introduction (line 69) "Here we hypothesize that the zebrafish innate immune system can be modulated by a tumor, itself capable of generating an immunosuppressive TME or subjected to elimination." The authors need to provide a more accurate and informative title.

We thank reviewer #3 for this comment and have changed the title to:
"Innate immune evasion revealed in the zebrafish xenograft model"

Minor

1) Line 236; Using the runx/panther mutants is an elegant approach to looking at the effects of neutrophil/macrophage depletion, but these are definitely indirect, rather than direct tests of these hypotheses; both mutations may be having many other effects that alter the biology of the TME, most notably changes in MMP expression and/or activation. The issue of differences between host and tumour MMP suite has been recently reviewed (see Wyatt, et al. 2017 "Zebrafish Xenograft: An Evolutionary Experiment in Tumour Biology." Genes, 8(9), 220. <http://doi.org/10.3390/genes8090220> - it might be worth **adding this reference**). This is also relevant WRT the issue of using the zebrafish, vs. murine xenograft models; it was very gratifying to see this issue addressed directly and conclusively in this study.

We thank reviewer #3 for his comment and now include this reference.

2) Imaging - What lens (what NA)? The 5 μm step size seems awfully coarse; was PH diameter increased such that the FWHM was $> 5 \mu\text{m}$? Was this necessary due to extremely weak signals? Given that these were fixed specimens, I don't understand why a greater sampling frequency (smaller step size) was not used? This probably dramatically under-sampled the axial resolution of the system. I'm not certain, but the fact that the data therefore represents (possibly non-random) subsamples of the experimental populations may invalidate some of the assumptions of the statistical analyses.

We used a 25x water objective with a NA of 0.8. The choice to sample every 5 microns in z is due to the fact that the injected mammalian tumoral cells have a diameter on average 10 microns. Therefore, sampling every 5 microns we ensure to image every cell at least twice. This allow us to monitor and count all the cells of the tumor, layer by layer. Thus, we do not have a subsamples, each dot in a graph is the quantification from each xenograft in the confocal of a tumor from up to bottom.

We could have sampled at higher frequency of Z- however would create bigger images, longer acquisitions, more costs and that would only produce redundant information.

3) The authors should provide a spreadsheet with all 419 DEGs as a supplemental file; I am very curious about the specific genes you identified.

Please find Supplementary Table 1 with the DEGs.

Trivial

1) Minor English language issues throughout; needs proof reading. We thank reviewer #3 and have now did a more thorough proof reading.

2) figure 2f and comparable figures: y-axis "tumour size" is presumably tumour volume? Add units (presumably μm^3).

We apologize if it was not clear, but we quantified the number of tumor cells present in the tumor mass by confocal microscopy.

To assess tumor size, three representative slices of the tumor, from the top, middle and bottom, per z-stack per xenograft were analyzed and a proxy of total cell number of the entire tumor (DAPI nuclei) was estimated as follows:

Average (3 slices Zfirst + Zmidle +Zlast) \times total number slices/1.5

This constant was defined empirically by counting all slices in 50 tumors and then finding a constant. By counting all slices, we observed, as expected, that in every other slice we found that 50% of the cells were already quantified in the previous slice.

3) Finally, just as a comment, the experiment comparing the engraftment of 'SW480zEscapers' is very similar to one that I've had in mind for a few years (but have not been able to find the resources to do); it would be very interesting to compare the transcriptome of escapers to the parental SW480 cells; especially after two or more rounds of xenografting and recovery.

We thank reviewer#3 for the suggestion and indeed we performed a similar experiment to find the escapers clones. However, instead of several rounds of xenografting and recovering (which would need *in vitro* expansion, and therefore may alter phenotypes) we performed single cell RNAseq of SW480 xenografts at 2 time points: 1dpi and 4dpi. From 1dpi to 4dpi, some tumors disappear (cleared) and others remain (escapers). In this way we were able to identify the clones that get cleared (are no longer represented at 4dpi) and also the ones that remain and expanded.

Please see new Figures 8 and 9 and Supplementary Figure 7 and 8.

Similarly, the authors allude to the 'amazing plasticity and diversity of macrophage populations' in particular; it would be very interesting to use the zebrafish xenograft model to compare the transcriptomes of the tumour-permissive macrophages associated with progressing tumours to those associated with tumour clearance.

We thank reviewer#3 for this very interesting suggestion but unfortunately, we could not address this at the moment.

I look forward to seeing this, and the future work from this group published.

Thank you so much for your kind words.

REVIEWER COMMENTS

Reviewer #1 (Remarks to the Author):

The author's responses and revised manuscript robustly address all of the concerns raised by this reviewer in the initial review process.

Reviewer #2 (Remarks to the Author):

The authors have largely addressed my concerns. The addition of the scRNA-Seq experiment has improved the manuscript, providing evidence of immunoediting that was lacking in the original draft.

I also agree with the title change with the deemphasis on human tumours.

The authors are to be commended for their excellent work.

There remains a large number of minor grammatical errors, especially in the reworked sections, but these can be corrected during publication.

Reviewer #3 (Remarks to the Author):

The authors have adequately addressed all of my concerns, and I have no further suggestions for the improvement of this manuscript.

It was a pleasure to review this submission and I look forward to seeing future developments from this research.

Reviewer #4 (Remarks to the Author): with expertise in scRNA seq

In this study, Pova et al. studied the interaction between the innate immune system and human cancer cells using the zebrafish xenograft model. They showed very convincingly that cells isolated from the same patient, SW480 and SW620, show different engraftment rates which is influenced by the interaction with the innate immune system, specifically macrophages and neutrophils. The observation that the engraftment is increased under radiotherapy and chemotherapy treatments further supports that the immune system is playing an important role. Experiments with mixed populations with different ratio, implicate a role in immunoediting. Overall, besides the following concerns which mostly focus on the single cell RNA-Seq analysis and statistical aspects, the manuscript is written well and has important implication on studying the role of innate immune system in tumor engraftment utilizing the zebrafish xenograft system. I am looking forward to seeing this paper published in Nature communication.

Major concerns

1. Single cell RNA-Seq analysis:

a. Quality controls graphs are missing. For each time point, number of UMIs per cell, % mitochondrial genes, % ribosomal gene should be shown before and after filtering to get an

estimate on the quality of the data.

b. Since the UMAP doesn't show a clear separation of the five clusters the decision to split to 5 clusters should be better supported by several additional analysis:

i. For this number of cells, UMAP doesn't usually visualize the clusters well. Can we see the cluster embedding on top of PCA or MDS (multidimensional scaling) which is more suited for this number of cells?

ii. Silhouette analysis to support the choice of 5 clusters.

iii. Heatmap representing the marker genes for each cluster showing that each cluster has a unique transcriptional program.

iv. Comparison analysis of the clusters to known published datasets (for example: Dai et al 2019).

c. The use of the word 'clone' in the context of single cell RNA-Seq is confusing. Clones refers to genetic difference and not transcriptional. If you want to show that these are unique clones you can either infer copy number variation from the scRNA-Seq data or do WGS. Otherwise, I would avoid the word 'clones' and replace it with transcriptional states or cell states.

2. Since you have this unique single cell RNA-Seq dataset it will be interesting to see if and how each cluster changes over the time of the tumor engraftment.

3. Figure 4g – the statistical test performed in this analysis is problematic since M1-like and M2-like are dependent ($M1+M2=100\%$). A better option would be to do the statistical tests on the raw numbers and not the percentage to reflect significant changes in M1 and M2-like. In supplementary figure 4 I can see the raw numbers, but the comparison is different so it's hard to tell if these are significant.

4. Thresholds should be consistent throughout the paper. Different FDR thresholds are used over the paper (For example: Method/RNA-Seq analysis uses $FDR < 0.05$ and Pathway Enrichment Analysis of a ranked gene list using GSEA uses $FDR < 0.07$).

5. Method/RNA-Seq analysis – \log_2 fold change threshold should be greater than 1. Traditionally, 1.5 or 2 is used.

Minor concerns:

6. What do you mean by 'Normalized Expression'? is this z-score?

7. The P values across the paper are not indicated if they were corrected for multiple hypothesis testing. In addition, it will be very helpful to show effect size in addition to P-value since it will give us the extent of the difference (for example, Cohen's d).

We would like to thank again all reviewers for the critical and careful reading of our manuscript and the opportunity to address all concerns raised by Reviewer#4, improving greatly our manuscript.

REVIEWER COMMENTS

Reviewer #1 (Remarks to the Author):

The author's responses and revised manuscript robustly address all of the concerns raised by this reviewer in the initial review process.

Reviewer #2 (Remarks to the Author):

The authors have largely addressed my concerns. The addition of the scRNA-Seq experiment has improved the manuscript, providing evidence of immunoediting that was lacking in the original draft.

I also agree with the title change with the deemphasis on human tumours.

The authors are to be commended for their excellent work.

There remains a large number of minor grammatical errors, especially in the reworked sections, but these can be corrected during publication.

Reviewer #3 (Remarks to the Author):

The authors have adequately addressed all of my concerns, and I have no further suggestions for the improvement of this manuscript.

It was a pleasure to review this submission and I look forward to seeing future developments from this research.

Reviewer #4 (Remarks to the Author): with expertise in scRNA seq. In this study, Pova et al. studied the interaction between the innate immune system and human cancer cells using the zebrafish xenograft model. They showed very convincingly that cells isolated from the same patient, SW480 and SW620, show different engraftment rates which is influenced by the interaction with the innate immune system, specifically macrophages and neutrophils. The observation that the engraftment is increased under radiotherapy and chemotherapy treatments further supports that the immune system is playing an important role. Experiments with mixed populations with different ratio, implicate a role in immunoediting. Overall, besides the following concerns which mostly focus on the single cell RNA-Seq analysis and statistical aspects, the manuscript is written well and has important implication on studying the role of innate immune system in tumor engraftment utilizing the zebrafish xenograft system. I am looking forward to seeing this paper published in Nature communication.

1. Single cell RNA-Seq analysis: a. Quality controls graphs are missing. For each time point, number of UMIs per cell, % mitochondrial genes, % ribosomal gene should be shown before and after filtering to get an estimate on the quality of the data.

We thank reviewer #4 for pointing this out. We have now added a supplementary figure, with the number of genes, number of UMI and percentage of UMIs attributed to mitochondrial genes, divided by library (**new Supplementary Figure 8a**). Overall, all libraries are mostly similar in their properties. A minority of cells in the libraries belonging to 4dpi seem to have a predominance of UMIs in mitochondrial genes. This suggests the presence of dying cells, which is expected at 4dpi time point, since we had less fish with tumors, we had to dilute the sample and therefore the sorting process was less efficient, and cells had to wait on ice. Nonetheless, after filtering, most of these cases have been removed.

b. Since the UMAP doesn't show a clear separation of the five clusters the decision to split to 5 clusters should be better supported by several additional analysis: i. For this number of cells, UMAP doesn't usually visualize the clusters well. Can we see the cluster embedding on top of PCA or MDS (multidimensional scaling) which is more suited for this number of cells?

We agree that a PCA is likely a fairer representation of the data, therefore we include a new supplementary figure (**new Supplementary Figure 8b**) with a PCA representation of the data. Like the UMAP, it does not suggest very clearly separable clusters among our cells. Considering the top 3 Principal Components (see **Figure R1a-c**), the chosen six clusters have distinguishable gene expression profiles, a similar message as the UMAP shows in a single 2D plot. Although the UMAP may be more suited for a greater number of cells, it should be a reasonable approach to make a 2D representation of the similarities between our ~800 cells, and in our opinion does a good job generating a compact summary of our data. Importantly, we think it is relevant to mention that all these cells come from a single human cancer cell line (SW480 cells). Therefore, we do not expect to see many different cell types, as we would see in a complex tissue sample such as from a cancer patient- where you can find not only epithelial cancer cells- different cell types, clones and subclones but also all the different cells present in the TME like T cell, B cells, myeloid cells, endothelial cells and stromal cells (*Li, et al. Nat Genet* <https://doi.org/10.1038/ng.3818>).

Figure R1 | a. Elbow plot indicating the variance explained by the top 20 principal components; b-c. Principal Component Analysis (PCA) plots of PC1 versus PC2 (a) and PC2 versus PC3 (b), colored by cell cluster and divided by time point.

ii. Silhouette analysis to support the choice of 5 clusters.

A silhouette analysis reveals a relatively low average silhouette width of 0.01 for our 6 clusters (**Figure R2a**). Nonetheless, any other choice of cluster number that we tried revealed an equally low average silhouette width of around -0.05 and 0.05 (**Figure R2b**). Even two clusters, which seem to suggest a higher average score, is clearly dominated by a single cluster, making it a moot choice. Therefore, we do not think this is a very useful criteria to choose number of clusters. One could also argue that then we should not try to cluster cells at all. This is a valid point, and it is important to mention that the main messages in our work do not depend on specific definitions of clusters.

Figure R2 | **a.** Silhouette plot of the 6 clusters used in the paper (cluster0=1, ..., cluster5=6) based on the Euclidean distance obtained using the scaled expression values of the 2000 most varying genes; **b.** Average and one standard deviation intervals for the silhouette value of all cells, for different cluster numbers, inferred using the partition around medoids (pam) function of the cluster R package (a robust version of k-means clustering, with $k=2, \dots, k=10$) based on the scaled expression values of the 2000 most varying genes.

To emphasize this, we provide a supplementary figure (**new Supplementary Figure 8d**) with the pathway enrichment scores depending not on clusters but on time point, where we can still see a decrease of notch activation and MYD88 independent TLR4 cascade at 4dpi, among other patterns that we state in the paper. The number of clusters was indirectly chosen using the resolution parameter in the FindClusters function from Seurat, where we use the default Louvain algorithm to optimize module detection in a K-nearest neighbor graph, as implemented in Seurat. We were nonetheless fairly conservative in our parameter choice (we used a resolution of 0.5) as we felt the presence of more clusters was too arbitrary and not informative.

We agree that the choice of 6 clusters is somewhat arbitrary, but these clusters, chosen using a conservative parameter with a proven and widely used algorithm, contain groups of cells with similar expression profiles, enabling a more finegrained analysis than just using the two time points.

iii. Heatmap representing the marker genes for each cluster showing that each cluster has a unique transcriptional program.

We now provide as supplemental figure a heatmap with all significant markers (**new Supplementary Figure 8c**), providing further evidence that cells in each cluster have distinct gene expression features with potential biological relevance.

iv. Comparison analysis of the clusters to known published datasets (for example: Dai et al 2019).

We thank the reviewer#4 for the suggestion. Dai et al 2019 only analyze one patient and do not look for relevant biological signatures in terms of intestine biology (stem cells/differentiation states). Therefore, we used instead the signatures present in *Li, et al. Nat Genet* <https://doi.org/10.1038/ng.3818>. Li et al, 2017 compare 11 primary CRC primary tumors with normal mucosa and search for the stem cells / differentiation states. We searched for the same markers and as expected from our previous analysis the stem cell / TA markers are indeed the more abundant. The markers that we have selected based on previous literature of the development of the gut are also present in this paper (**Figure R3**). We did go into a bit more detail because Notch and Wnt signaling clearly popped out, so we were able to go deeper into the “cell fate decisions between these cells”.

Figure R3| Heatmap of normalized expression values of genes from Figure 3 of Li et al. (2017). Some genes are not represented because they were not expressed in our cells. Colors represent expression values scaled by row (Z-scores). Genes marked with squares are genes we already include in other supplementary figures in the text.

c. The use of the word 'clone' in the context of single cell RNA-Seq is confusing. Clones refers to genetic difference and not transcriptional. If you want to show that these are unique clones you can either infer copy number variation from the scRNA-Seq data or do WGS. Otherwise, I would avoid the word 'clones' and replace it with transcriptional states or cell states.

We thank reviewer#4 for the suggestions and have now changed to "cell states" or sometimes subclones.

2. Since you have this unique single cell RNA-Seq dataset it will be interesting to see if and how each cluster changes over the time of the tumor engraftment.

We thank the reviewer#4 for this interesting and relevant suggestion. Nonetheless, particularly considering the relative similarity of the clusters, we wish to make no further claims other than the change in the relative proportions of each cluster (and their specific markers) between the two time points. Moreover, the clusters themselves already capture most of the change we want to explore, as eg. cluster 3, 4 and 5 have almost all cells from a single time point.

3. Figure 4g – the statistical test performed in this analysis is problematic since M1-like and M2-like are dependent ($M1+M2=100\%$). A better option would be to do the statistical tests on the raw numbers and not the percentage to reflect significant changes in M1 and M2-like. In supplementary figure 4 I can see the raw numbers, but the comparison is different so it's hard to tell if these are significant.

We thank reviewer#4 for the comment we now present a new graph with absolute numbers and show the statistical analysis in raw numbers (**see new Figure 4**). We maintained the % graph since is more visual.

4. Thresholds should be consistent throughout the paper. Different FDR thresholds are used over the paper (For example: Method/RNA-Seq analysis uses $FDR < 0.05$ and Pathway Enrichment Analysis of a ranked gene list using GSEA uses $FDR < 0.07$).

We used $FDR < 0.07$ in the bulk analysis because we wanted to highlight the “allograft rejection pathway and the IL6/STAT3 pathways” that were close-to-significant and relevant for our manuscript flow. We could change the subsequent analysis to $FDR < 0.07$ just for consistency but we believe this is not a major concern since would reduce the stringency of the rest of the analysis.

5. Method/RNA-Seq analysis – \log_2 fold change threshold should be greater than 1. Traditionally, 1.5 or 2 is used.

To our knowledge the fold change cut-off most often used is 1.5 or 2, which will correspond to \log_2FC of 0.58 or 1. In this case, we are using a \log_2FC of 1, i.e. corresponding to a fold change of 2, which we believe is strict enough.

Minor concerns:

6. What do you mean by ‘Normalized Expression’? is this z-score?

We thank the reviewer#4 for pointing out a lack of clarity in the description. We have tried to clarify this in the methods section by adding an extra sentence: "The Normalized expression values of genes used for visualization correspond to the \log_2 of total cell counts divided by 10000 (very similar to the traditional CPM). In the case of heatmaps, values are scaled by row."

7. The P values across the paper are not indicted if they were corrected for multiple hypothesis testing. In addition, it will be very helpful to show effect size in addition to P-value since it will give us the extent of the difference (for example, Cohen’s d).

We thank reviewer #4 for the comment. All p-values presented in the text, namely those regarding transcriptomics analysis, are all corrected for multiple hypothesis testing.

Additionally, for small number of samples (< 10), we performed an effect size analysis – Cohen’s D with a Hedges’ g correction (g), using Cohen’s D 1988 scale: $g = 0.2$ Low; $g = 0.5$ Moderate; $g = 0.8$ High.

REVIEWERS' COMMENTS

Reviewer #4 (Remarks to the Author):

My concerns are addressed in the revised manuscript.